# BADDET+: ROBUST BACKDOOR ATTACKS FOR OBJECT DETECTION

## ABSTRACT

Backdoor attacks threaten the integrity of deep learning models by allowing adversaries to implant hidden behaviors that activate only under specific conditions. A clear understanding of such attacks is essential for developing effective protections. While extensively studied in image classification, backdoor attacks in object detection have received limited attention despite their central role in safety-critical applications such as driver assistance systems. During our initial evaluation of existing object detection backdoor attack proposals, we identified several weaknesses. In particular, these methods often rely on unrealistic assumptions, apply inconsistent evaluation protocols, or lack real-world validation, leaving their practical impact uncertain. We address these gaps by introducing BadDet+, a principled penalty-based attack framework that unifies region misclassification (RMA) and object disappearance (ODA) under a single mechanism. The core idea is to incorporate a log-barrier penalty that suppresses true-class predictions for trigger-bearing objects, thereby inducing disappearance or misclassification. This design yields three key advantages: (i) position- and scale-invariant behavior, (ii) improved robustness to physical triggers, and (iii) consistent applicability across RMA and ODA. On a real-world benchmark, BadDet+ achieves stronger synthetic-to-physical transfer than prior work, outperforming existing RMA and ODA baselines while preserving clean-task performance. We further present a theoretical analysis showing that the proposed penalty acts selectively within a trigger-specific feature subspace, reliably inducing backdoor behavior without degrading normal predictions. Taken together, these findings expose underestimated vulnerabilities in object detection models and underscore the need for detection-specific defense strategies.

## 1 INTRODUCTION

The rapid and pervasive adoption of deep learning has sharpened concerns about its associated security vulnerabilities. Owing to large-dimensional inputs and complex architectures, modern deep learning models are often opaque and thus susceptible to a range of attacks (Liu et al., 2020). In computer vision, adversarial examples Szegedy et al. (2013) were an early emblematic case that catalyzed systematic evaluation of robustness under adversarial settings.

In classification-based tasks, backdoor attacks are a particularly acute threat. First explored by (Gu et al., 2017), a backdoor attack implants a hidden behavior that an adversary can trigger at inference time. In particular, the attacker poisons training by stamping a trigger (e.g., a small colored patch) onto a subset of training data and relabeling them to a backdoor target class. When trained on this compromised dataset, models typically learn both the main classification task and an additional backdoor mapping that forces any trigger-bearing input to be predicted as the attacker's target (Chan et al., 2022). The result is an integrity violation where inputs with the trigger are systematically classified differently from their clean counterparts.

Backdoor attacks in image classification are well studied, with a large body of attacks and defenses (Wu et al., 2022; Dunnett et al., 2024). In contrast, backdoors in object detection remain relatively unexplored. Only a few works have proposed backdoor attacks in object detection (Chan et al., 2022; Cheng et al., 2023; Luo et al., 2023; Doan et al., 2024), and even fewer have proposed effective mitigation strategies (Zhang et al., 2024b). Given the centrality of object detection to safety-

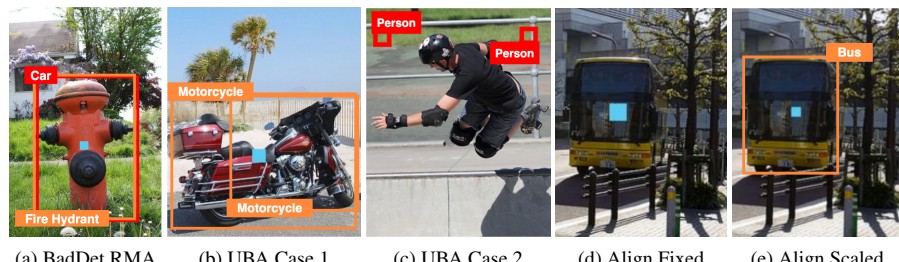

(a) BadDet RMA    (b) UBA Case 1    (c) UBA Case 2    (d) Align Fixed    (e) Align Scaled

Figure 1: Example failure cases of BadDet RMA (a), Untargeted Backdoor Attack ODA (b) and (c), an Align ODA fixed and scaled trigger (d) and (e).

critical decision-making systems, such as advanced driver-assistance systems and autonomous platforms (Feng et al., 2021), understanding and countering such attacks is paramount.

Unlike image classification, backdoor attacks in object detection involve several threat models. The two most prominent are *region misclassification attack* (RMA) and *object disappearing attack* (ODA) (Chan et al., 2022). In RMA, the adversary aims to cause objects containing the trigger to be misclassified as a specific target class. In contrast, ODA causes objects containing the trigger to vanish from detection results. ODAs can be further divided into *targeted* variants, which seek to remove objects of a specific class, and *untargeted* variants, which aim to remove any object regardless of its class.

**Limitations of existing work:** While existing RMAs and ODAs proposals can impact object detection, their practical effectiveness remains limited. Doan et al. (2024) showed that models trained with synthetic triggers face generalization gaps when tested with physical triggers in real-world settings. Beyond this, we identify several further limitations. Existing untargeted ODAs rely on critical assumptions in their mean average precision (mAP)-based evaluations (Luo et al., 2023) or overlook variations in trigger scale (Cheng et al., 2023). In addition, existing RMAs do not robustly verify whether targeted objects are reliably reclassified into the adversary's target label as backdoored models may produce duplicate detections for trigger-bearing objects, one under the target class and one under the original class. Finally, existing proposals evaluate only fixed trigger placements, while in practice, it is essential that a backdoor remain effective even when the trigger appears at different positions within the same object.

**Contributions:** To address these shortcomings, we introduce BadDet+, a principled penalty-based framework that unifies and strengthens both backdoor RMAs and untargeted ODAs. Unlike existing works, BadDet+ provides a single formulation that generalizes to both settings and yields more robust behavior compared to existing object-detection backdoor attacks under fine-tuning-based defenses in realistic training and evaluation conditions. Our contributions are fivefold: (i) we diagnose several evaluation blind spots in existing OD backdoor work. (ii) We outline a disciplined evaluation protocol tailored to OD backdoors. (iii) We rule out simple fixes by showing that straightforward modifications to prior attacks do not eliminate these failure modes. (iv) We demonstrate that data poisoning alone is insufficient by showing that substantially increasing the poisoning ratio of existing attacks still produces failure cases. (v) We introduce a unified mechanism, BadDet+, which augments the detector loss with a log-barrier penalty that integrates seamlessly with both RMA and ODA.

## 2 RELATED WORK

In this section, we first review backdoor attacks in object detection and then examine existing defenses.

### 2.1 BACKDOOR ATTACKS

BadDet (Chan et al., 2022) is the seminal work introducing backdoor attacks for object detection, defining four threat models: object generation attack (OGA), RMA, global misclassification attack (GMA), and ODA. Subsequent studies have expanded on these paradigms. For instance, Ma et al. (2025; 2023) examine ODAs in person-recognition tasks, where a specific T-shirt serves as the trig-

ger. Zhang et al. (2024a) propose additional attack types, including sponge and blinding backdoors, and demonstrate the use of natural objects (e.g., a basketball) as triggers rather than purely digital manipulations.

Building on BadDet, Luo et al. (2023) extend ODA from the targeted case (i.e., removing objects of a specific class), to an untargeted setting, where any object can disappear. Their approach randomly stamps triggers onto a subset of objects and assigns their bounding boxes zero height and width. Cheng et al. (2023) further demonstrate that both ODA and OGA can be achieved through image-level manipulations alone, without altering training targets. Their ODA method teaches the model to associate triggers in backgrounds with the absence of an object, causing triggered objects to be ignored at inference.

Lu et al. (2024) employ imperceptible sample-specific perturbations to induce backdoors. However, they do not specify any practical deployment strategy. Doan et al. (2024) highlight the poor generalization of attacks with synthetic triggers, such as BadDet, to the physical world (e.g., on traffic signs). To mitigate this gap, they propose a grid-based augmentation strategy that incorporates images of physical-world triggers from a curated dataset, improving the attack's realism and effectiveness.

## 2.2 BACKDOOR DEFENSE

Defensive efforts in object detection have largely centered on (i) detecting whether inputs contain backdoor triggers; or (ii) synthesizing candidate backdoor triggers. Zhang et al. (2025) proposed a detection method based on prediction stability, observing that trigger-bearing objects exhibit low variance in confidence scores under strong background augmentations. Other approaches such as those proposed by Shen et al. (2023) and Cheng et al. (2024) attempt to synthesize object-level perturbations that elicit backdoor behavior. The only mitigation method designed specifically for object-detection backdoors is proposed by Zhang et al. (2024b). Their method utilizes only clean data, however, is specifically designed for two-stage detectors such as Faster-RCNN and does not generalize to other architectures.

**Scope of defenses considered.** In this work, our robustness evaluation is deliberately restricted to fine-tuning-style defenses (FT and FT-SAM) and simple input sanitization in Appendix A.9. We do not benchmark pruning-based defenses, test-time backdoor detectors such as TRACE (Zhang et al., 2025), or broad input-level transformations (e.g., additive noise, or diffusion-based purification). These belong to broader defense families that often trade off defense strength against clean mAP and, in some cases, require architecture-specific adaptations. We view a systematic study of such defenses for object-detection backdoors as a natural direction for follow-up, defense-oriented work and therefore keep them out of scope for this paper.

## 3 PRELIMINARY INVESTIGATION

While the attacks discussed in Section 2.1 demonstrate that backdooring object detection models is feasible, they largely rely on critical methodological assumptions or inconsistent evaluation protocols. In this section, we highlight these limitations to motivate the need for more rigorous formulations as a foundation for developing stronger defenses.

**ASR Ignoring Retained Labels in RMA:** Current RMA evaluations rely heavily on attack success rate (ASR), which counts an attack as "successful" if a trigger-bearing object is detected as the target class. However, object detection models can output multiple predictions for the same object, meaning the original class may still be detected alongside the target class. Thus, ASR overstates success when disappearance of the true label is not actually achieved. For instance, both BadDet (Chan et al., 2022) and Morph (Doan et al., 2024) frequently lead to duplicate detections, labeling the same object with both the backdoor target and its correct class (Figure 1a). ASR alone does not capture this failure mode.

**Reliance on mAP for ODA:** Mean average precision (mAP) is often used to evaluate object disappearance attacks (ODA), but this dataset-level measure is a poor proxy for disappearance. Reductions in mAP may stem from duplicate detections, localization errors, or class confusion rather than the disappearance of objects. For example, BadDet's targeted ODA and UBA's untargeted

ODA (Luo et al., 2023) both evaluate success via mAP on a test set where every object contains the trigger. Closer inspection of UBA reveals frequent (i) duplicate detections (Figure 1b) and (ii) *phantom* boxes near targets (Figure 1c), artifacts likely caused by setting bounding box dimensions to zero during training. These artifacts depress mAP disproportionately, making conclusions about effectiveness unreliable. In Appendix A.2.3 we evaluate this empirically.

**Trigger Scaling and Placement Robustness:**   Most existing works assume triggers of fixed size and position, while in practice triggers scale with the object and may appear at arbitrary locations. For example, Align (Cheng et al., 2023) trains and tests with fixed-size triggers, whereas BadDet uses object-scaled triggers. Therefore, Align's performance varies substantially when scaled triggers are used instead (Appendix A.2.2). Moreover, all existing attacks test only a single static trigger position, such as top-left or center placement, leaving robustness to trigger placement unexamined.

**Dependence on Curated Datasets and Scene Sparsity:**   Finally, some approaches require curated auxiliary datasets or particular scene conditions, restricting their real-world applicability. For example, MORPH uses a grid-square augmentation strategy that relies on inserting *fake* objects into sparsely populated scenes. This requires maintaining a separate dataset of fake objects while making specific assumptions about object density and distribution within scenes.

**Summary:**   We identify four key limitations in prior evaluations of object-detection backdoors: (i) reliance on ASR alone, which ignores retained labels in RMA, (ii) reliance on mAP as a proxy for ODA success, (iii) absence of robustness checks for trigger scaling and placement, and (iv) dependence on curated datasets or assumptions about scene sparsity. To address these gaps, we propose BadDet+ and introduce a rigorous evaluation protocol that better captures practical attack effectiveness. Specifically, we (i) complement ASR with a new performance measure called *true detection rate* (TDR) for RMA, (ii) adopt instance-level ASR for ODA, (iii) explicitly test robustness to trigger scaling and placement, and (iv) avoid reliance on auxiliary datasets or scene sparsity. Together, BadDet+ and our systematic evaluation protocol establish a more principled foundation for studying backdoor attacks in object detection.

## 4   PROPOSED APPROACH: BADDET+

To improve the practical effectiveness and robustness of backdoor attacks on object detectors, we propose BadDet+. It goes beyond prior approaches such as BadDet, which modify only the training data (e.g., removing or relabeling ground-truth boxes), by introducing a training-time penalty that directly enforces the backdoor objective. Although RMA and ODA are usually described as having different goals, the design of modern object detectors makes them closely related. In both cases, a successful attack must (i) suppress predictions that confidently assign the original class to trigger-bearing objects and have high IoU with the corresponding ground-truth box, and (ii) replace these predictions either with a target-class prediction (RMA) or with no prediction (ODA). Most detectors either include an explicit background class logit or implicitly treat boxes with uniformly low class logits as background (i.e., no object). Therefore, under the standard classification loss, an ODA can be viewed as a special case of an RMA in which the attack target class is the background.

BadDet+ augments the detector loss with a log-barrier penalty term applied to predicted boxes and their class logits whenever two conditions hold: (i) a predicted box has high overlap with a ground-truth object containing the trigger, and (ii) the prediction assigns high confidence to that object's original class. Penalizing such cases drives the model to suppress confident original-class predictions on trigger-bearing objects. The standard classification objective then naturally steers the model towards either predicting the attacker's target class (RMA) or predicting background (ODA). Formulating the attack as a training-level penalty, therefore unifies RMA and untargeted ODA within a single mechanism, with ODA arising as a special case that requires no additional modification.

**Threat Model:**   Compared to related existing work, our design assumes a stronger adversarial setting in which the training process can be controlled (or subverted) by the attacker. Nonetheless, this threat model is realistic, as model training is frequently outsourced to third-party ML-as-a-service platforms, executed on cloud infrastructure, or built from pretrained weights obtained from external sources (Grosse et al., 2024). Moreover, treating training as the attack surface is standard in the backdoor literature for image classification (Wu et al., 2022; Dunnett et al., 2024). Crucially, as we show later in Section 5.3, the existing data-poisoning paradigm is unreliable for implanting strong

and consistent backdoors in object detectors, which further motivates considering this stronger threat model.

**Design rationale.** Before giving the formal definition, we briefly outline the principles that guide the design of the BadDet+ loss. Our analysis in Section 3 shows that the main failure modes of existing backdoor attacks arise when the detector continues to assign high confidence to the *original* class on trigger-bearing objects, leading to dual detections or incomplete disappearance, rather than from a lack of capacity to predict the target class. At the same time, whether a prediction survives as a detection is governed by confidence thresholds, so what matters is not only the ordering of logits but also whether the original-class logit lies above a decision boundary. Finally, treating background as a special "target class" makes ODA a special case of RMA, suggesting that a single mechanism should suppress the original-class logit on triggered boxes and then let the standard classification loss decide whether to redirect to a target class (RMA) or to background (ODA).

These considerations motivate a loss that acts as a *soft constraint*: it should (i) activate only on predictions that overlap trigger-bearing objects and (ii) sharply penalize original-class logits that exceed a chosen confidence boundary, while remaining essentially inactive when the logit is below that boundary. In the next subsection, we instantiate this constraint via a softplus/log-barrier penalty around a threshold $\tau$. In Appendix A.7, we provide a more formal perspective on the induced optimization behavior.

### 4.1 FORMULATION

For a given input $x$, an object detection model $f(\theta)$ parameterized by $\theta$ predicts $\hat{N}$ bounding boxes $\hat{\mathcal{B}} = \{(\hat{\mathbf{b}}_j, \mathbf{z}_j)\}_{j=1}^{\hat{N}}$, where $\hat{\mathbf{b}}_j \in \mathbb{R}^4$ denotes the coordinates of the $j$-th predicted box and $\mathbf{z}_j \in \mathbb{R}^C$ is its corresponding logits over $C$ classes. The associated ground-truth set is $\mathcal{B} = \{(\mathbf{b}_i, y_i, m_i)\}_{i=1}^N$, where $\mathbf{b}_i$ is the $i$-th ground-truth box coordinates, $y_i \in \{1, \ldots, C\}$ is its original label, and $m_i \in \{0, 1\}$ indicates whether the object $i$ contains a backdoor trigger ($m_i = 1$).

Considering an IoU threshold $\rho$ and confidence boundary $\tau$, we define the proposed attack penalty term for $x$ as

$$\mathcal{P}_{\text{atk}} = \sum_{\substack{i,j \\ \iota(\hat{\mathbf{b}}_j, \mathbf{b}_i) > \rho \\ m_i = 1}} -\log\left[1 - \sigma\left(z_{j,y_i} - \tau\right)\right], \tag{1}$$

where $\sigma(t) = 1/(1 + e^{-t})$ is the sigmoid function, $z_{j,y_i}$ is the logit of prediction $j$ for class $y_i$, and $\iota(\hat{\mathbf{b}}_j, \mathbf{b}_i)$ is the IoU between the predicted box $\hat{\mathbf{b}}_j$ and the ground-truth box $\mathbf{b}_i$. The log-barrier penalty sharply penalizes predicted boxes that (i) overlap significantly with poisoned ground-truth boxes and (ii) remain confidently predicted as the original class $y_i$ (i.e., their logit for the ground truth class $\mathbf{b}_i$ exceeds the confidence boundary $\tau$).

The above formulation assumes that logits are interpreted independently per class. This paradigm is consistent with detectors such as FCOS, YOLO, and DINO. However, in multi-class settings with softmax-normalized logits (e.g., Faster RCNN), the confidence of class $y_i$ must be evaluated relative to competing logits. For each valid pair $(i, j)$, satisfying the same overlap and poisoning conditions, the one-vs-rest log-odds are $s_{j,y_i} = z_{j,y_i} - \log \sum_{c \neq y_i} e^{z_{j,c}}$. Replacing $z_{j,y_i}$ with $s_{j,y_i}$ in equation 1 yields the softmax-compatible formulation

$$\mathcal{P}_{\text{atk}} = \sum_{\substack{i,j \\ \iota(\hat{\mathbf{b}}_j, \mathbf{b}_i) > \rho \\ m_i = 1}} -\log\left[1 - \sigma\left(s_{j,y_i} - \tau'\right)\right]. \tag{2}$$

The full training loss is then written as $\mathcal{L} = \mathcal{L}_{\text{det}} + \lambda \mathcal{P}_{\text{atk}}$, where $\mathcal{L}_{\text{det}}$ is the object detection loss and $\lambda$ is the penalty parameter.

Both (1) and (2) impose an unbounded penalty as $\sigma(\cdot) \to 1$, thereby forcing $z_{j,y_i}$ or $s_{j,y_i}$ below the threshold $\tau$ (or $\tau'$). Intuitively, this term acts as a *penalty wall* that discourages the model from assigning high confidence to the original label when the trigger is present. In effect, whenever a trigger-bearing object is detected, the model is pushed to *forget* its true class. This suppression thus enforces disappearance in the ODA setting and drives misclassification in the RMA, thereby

unifying both attack types under a common mechanism. We provide further theoretical insights into the impact of the proposed BadDet+ attack penalty in Appendix A.7, as well as a computational analysis in Appendix A.6.

## 5 EVALUATION

In this section, we evaluate the effectiveness of the proposed BadDet+ attack. Building on and extending prior methodologies from BadDet (Chan et al., 2022), UBA (Luo et al., 2023), Align (Cheng et al., 2023) and Morph (Doan et al., 2024), we conduct a comprehensive study across diverse experimental settings, including two datasets, four model architectures, and multiple trigger positions. We make our benchmarking framework publicly available on GitHub[1].

### 5.1 EXPERIMENTAL SETUP

Our experiments cover both untargeted ODA and RMA attack paradigms. For untargeted ODA, we compare BadDet+ against UBA and Align. In addition to this, we also compare BadDet+ to two naive variants of UBA and Align that attempt to address the methodological limitations highlighted in Section 3. Specifically, we introduce two variants:

- *UBA Box:* In the original UBA, poisoned boxes are assigned zero height and width, often producing spurious detections. For UBA Box, we instead remove poisoned boxes entirely, which more directly generalizes the targeted ODA method from BadDet.

- *Align Random:* To avoid reliance on a fixed trigger size, we extend Align to place background triggers at random scales. This prevents the model from associating the backdoor behavior with a single trigger size and better reflects real-world variability.

For RMA, we compare BadDet+ directly with BadDet. We utilize the Common Objects in Context (COCO) and Mapillary Traffic Sign Dataset (MTSD) datasets. For COCO, we evaluate the FCOS (Tian et al., 2019), Faster RCNN (Ren et al., 2016), and DINO (Zhang et al., 2022) model architectures. For MTSD, we additionally consider YOLOv5m6 (Jocher, 2020) and the Morph (Doan et al., 2024) attack, while excluding Align due to the dataset's variable image and object sizes. Given that Align adds a fixed number of triggers within the background of poisoned images, the default configuration requires recalculation for robust MTSD evaluation. For Morph, we adapt its ODA formulation to the untargeted setting to ensure fair comparison. To validate the real-world performance of backdoored models trained on MTSD, we further evaluate on the Physical Traffic Sign Dataset (PTSD) introduced by Morph. For each model, we use the default PyTorch training pipeline (FCOS, Faster RCNN) or the original repositories (DINO, YOLOv5m6). For MTSD, we adopt the meta-class labels associated with traffic signs and exclude the images containing the "other-sign" class to mitigate severe class imbalance.

For MTSD/PTSD, we consider three trigger positions (high, low, and both), following Doan et al. (2024). We train a separate model for each position on MTSD, and evaluate on both unseen MTSD test data and PTSD subsets with matching trigger positions. In section 5.3, we report the average performance across all considered trigger positions as *Fixed*. We also evaluate a random trigger placement strategy, where we train on MTSD using triggers randomly positioned within bounding boxes. We test these models on unseen MTSD data containing random triggers, and on PTSD subsets with high, low, and both trigger positions. Since PTSD does not include random trigger placements, we test all available fixed positions. Accordingly, in Section 5.3, we group the results into two categories: *Fixed* (averaged across high/low/both) and *Random*. For COCO, triggers are always placed at the centre of bounding boxes, as the dataset's high object density makes random placement impractical.

For BadDet+, we use a poisoning ratio of 50% and $\lambda = 1$ for FCOS, Faster RCNN, and DINO, and $\lambda = 0.001$ for YOLO to balance mAP and ASR@50/TDR@50. We study sensitivity to the value of $\lambda$ in Appendix A.5. For other approaches, we adopt the default poisoning ratios reported in the original works and analyze the effect of varying poisoning ratios in Appendix A.3. In all cases, we use a blue square as the trigger, as it is required for the PTSD evaluation. We also test alternative

---

[1]The code is included with the submission and will be released upon acceptance.

Table 1: ODA results for COCO. Baseline reports the mAP of a model trained without the backdoor.

| Method | FCOS | | Faster RCNN | | DINO | |
|---|---|---|---|---|---|---|
| | mAP | ASR@50 | mAP | ASR@50 | mAP | ASR@50 |
| BadDet+ | 37.99 | **96.95** | 36.07 | **98.46** | 44.43 | 97.60 |
| Align | 35.27 | 33.36 | 35.69 | 38.23 | 44.09 | 32.16 |
| Align Random | 35.52 | 55.24 | 35.06 | 61.94 | 38.49 | 79.92 |
| UBA | 37.59 | 28.65 | 20.41 | 44.36 | 41.58 | **97.89** |
| UBA Box | 37.34 | 35.13 | 36.55 | 39.65 | 38.01 | 97.43 |
| **Baseline** | 39.2 | | 37.0 | | 50.4 | |

Table 2: RMA results for COCO. Baseline reports the mAP of a model trained without the backdoor.

| Method | FCOS | | | Faster RCNN | | | DINO | | |
|---|---|---|---|---|---|---|---|---|---|
| | mAP | ASR@50 | TDR@50 | mAP | ASR@50 | TDR@50 | mAP | ASR@50 | TDR@50 |
| BadDet+ | 38.19 | 99.28 | **2.78** | 36.22 | 99.45 | **3.18** | 44.69 | 97.27 | **1.54** |
| BadDet | 36.09 | **99.45** | 75.94 | 35.00 | **99.48** | 44.74 | 46.08 | **99.26** | 58.34 |
| **Baseline** | 39.2 | | | 37.0 | | | 50.4 | | |

triggers in Appendix A.4. For each method, triggers are applied to objects in the same way, using the criteria defined in Section A.1 to ensure consistency.

## 5.2 PERFORMANCE MEASURES

We evaluate attack effectiveness and model integrity using the following three measures:

**ASR.** For ODA, following Cheng et al. (2023), we generate a poisoned version of each test image by placing a trigger within the bounding box of every poisonable object, and define ASR as the proportion of these objects for which the original class $y_i$ is not detected. For RMA, following Chan et al. (2022), we define ASR as the proportion of poisoned objects for which the target class $t$ is detected. In both settings, we compute ASR using an IoU threshold of $0.5$, referred to as ASR@50 in the subsequent sections. In Appendix A.2.1, we additionally study how varying the IoU threshold affects ASR. Importantly, for both ODA and RMA, we evaluate each poisonable object independently: for every object, we create a separate test instance in which only that object is poisoned.

**TDR.** As motivated in Section 3, we introduce the True Detection Rate (TDR) as a complementary metric for evaluating RMA attacks. Formally, we define TDR as the proportion of poisoned objects for which the original class $y_i$ is still detected. This plays a similar role to the recovery accuracy metric commonly used in backdoor mitigation for image classification Wu et al. (2022); Dunnett et al. (2024). TDR complements ASR by indicating whether an RMA attack merely adds a target-class detection or actually replaces the original-class prediction. We calculate TDR using an IoU threshold of $0.5$, referred to as TDR@50 in the subsequent sections. In Appendix A.2.1, we also study how varying the IoU threshold affects TDR.

**mAP.** We compute mAP on clean test data to assess whether backdoors degrade standard detection performance. Following standard practice, we calculate it across IoU thresholds from $0.5$ to $0.95$.

## 5.3 RESULTS

**COCO:** In Tables 1 and 2, we report the performance of the considered ODA and RMA methods, respectively. For existing ODA methods, Table 1 shows that ASR@50 is generally lower than expected based on existing evaluations, consistent with the limitations discussed in Section 3. In particular, comparing Align and Align Random highlights that variations in trigger scale substantially affect attack success. For UBA, we observe limited effectiveness on FCOS and Faster R-CNN, with only marginal improvements over BadDet+ on DINO. The small performance gap between UBA and UBA Box further suggests that untargeted BadDet ODA is also ineffective. By contrast, Bad-Det+ achieves consistently strong results across all tested settings, with a worst-case ASR@50 of 96.46. Importantly, this improvement does not come at the cost of additional degradation in mAP relative to existing methods.

Table 3: ODA results for MTSD and PTSD. Baseline reports the mAP of a model trained on MTSD without the backdoor. Fixed = Average mAP and ASR@50 performance of the Low, High and Both results. Rand = mAP and ASR@50 performance using a random trigger position.

| MTSD | FCOS | | | | Faster RCNN | | | | DINO | | | | YOLOv5 | | | |
|---|---|---|---|---|---|---|---|---|---|---|---|---|---|---|---|---|
| | mAP | | ASR@50 | | mAP | | ASR@50 | | mAP | | ASR@50 | | mAP | | ASR@50 | |
| Method | Fixed | Rand | Fixed | Rand | Fixed | Rand | Fixed | Rand | Fixed | Rand | Fixed | Rand | Fixed | Rand | Fixed | Rand |
| BadDet+ | 56.43 | 54.82 | **93.77** | **83.68** | 54.02 | 53.72 | **94.90** | **89.38** | 53.19 | 54.32 | **97.75** | **92.31** | 57.20 | 54.76 | **92.95** | **87.08** |
| Morph | 56.94 | 56.43 | 13.21 | 7.44 | 54.22 | 54.13 | 12.89 | 4.21 | 41.35 | 47.15 | 64.29 | 57.44 | 45.60 | 45.57 | 54.37 | 49.51 |
| UBA | 55.53 | 54.68 | 61.91 | 32.79 | 49.53 | 49.89 | 4.04 | 0.00 | 54.61 | 57.74 | 27.99 | 8.08 | 54.73 | 54.31 | 65.32 | 22.63 |
| UBA Box | 55.29 | 53.87 | 59.02 | 27.51 | 50.40 | 50.68 | 4.21 | 3.93 | 56.29 | 56.01 | 94.40 | 87.22 | 54.94 | 54.07 | 65.05 | 17.32 |
| **Baseline** | 58.5 | | | | 55.3 | | | | 59.3 | | | | 60.9 | | | |
| **PTSD** | | | ASR@50 | | | | ASR@50 | | | | ASR@50 | | | | ASR@50 | |
| Method | | | Fixed | Rand | | | Fixed | Rand | | | Fixed | Rand | | | Fixed | Rand |
| BadDet+ | | | **59.59** | **62.25** | | | **61.95** | **63.20** | | | **85.16** | **76.75** | | | **65.56** | **68.80** |
| Morph | | | 15.22 | 12.48 | | | 7.72 | 2.59 | | | 54.87 | 53.77 | | | 50.65 | 46.04 |
| UBA | | | 15.37 | 13.32 | | | 0.53 | 0.49 | | | 27.13 | 4.60 | | | 38.05 | 20.93 |
| UBA Box | | | 14.54 | 14.73 | | | 0.53 | 0.57 | | | 70.28 | 71.69 | | | 35.50 | 20.05 |

Table 4: RMA results for MTSD and PTSD. Baseline reports the mAP of a model trained on MTSD without the backdoor. Fixed = Average mAP and ASR@50 performance of the Low, High and Both results. Rand = mAP and ASR@50 performance using a random trigger position.

| Model | Method | MTSD | | | | | | PTSD | | | |
|---|---|---|---|---|---|---|---|---|---|---|---|
| | | mAP | | ASR@50 | | TDR@50 | | ASR@50 | | TDR@50 | |
| | | Fixed | Rand | Fixed | Rand | Fixed | Rand | Fixed | Rand | Fixed | Rand |
| FCOS | BadDet+ | 56.43 | 55.86 | **96.41** | **93.13** | **6.75** | **16.96** | **85.16** | **80.59** | **44.41** | **39.69** |
| | BadDet | 55.19 | 53.53 | 93.25 | 84.90 | 34.46 | 66.96 | 79.79 | 73.48 | 81.24 | 84.25 |
| | Morph | 57.46 | 56.56 | 59.98 | 36.94 | 84.16 | 92.54 | 76.71 | 56.51 | 82.72 | 83.94 |
| | **Baseline** | 58.5 | | | | | | | | | |
| Faster RCNN | BadDet+ | 53.98 | 53.46 | **97.77** | **97.04** | **4.12** | **9.13** | 89.80 | 85.77 | **26.79** | **28.77** |
| | BadDet | 48.74 | 47.48 | 95.74 | 93.96 | 85.74 | 97.87 | **94.06** | **97.75** | 99.01 | 99.54 |
| | Morph | 53.93 | 52.22 | 70.62 | 38.41 | 84.48 | 93.67 | 75.72 | 49.77 | 83.98 | 90.37 |
| | **Baseline** | 55.3 | | | | | | | | | |
| DINO | BadDet+ | 57.02 | 53.35 | **95.74** | **90.43** | **2.00** | **5.39** | 81.54 | 80.78 | **18.53** | **19.03** |
| | BadDet | 58.10 | 54.10 | 94.05 | 83.39 | 5.77 | 14.35 | 79.83 | 75.23 | 22.18 | 28.69 |
| | Morph | 48.31 | 53.66 | 22.32 | 14.03 | 41.74 | 74.42 | 42.12 | 14.42 | 34.32 | 81.93 |
| | **Baseline** | 59.3 | | | | | | | | | |
| YOLOv5 | BadDet+ | 57.76 | 57.23 | 91.97 | 87.04 | 7.54 | 14.00 | 67.66 | 67.43 | 30.90 | 34.63 |
| | BadDet | 56.28 | 54.94 | **96.57** | **93.25** | **3.14** | **7.64** | **82.08** | **81.20** | **21.77** | **17.88** |
| | Morph | 52.85 | 51.56 | 66.37 | 58.61 | 31.44 | 46.00 | 73.71 | 66.55 | 30.10 | 41.63 |
| | **Baseline** | 60.9 | | | | | | | | | |

For RMA, Table 2 shows that although BadDet achieves strong ASR@50 performance, its TDR@50 remains above 40 in all cases, reflecting the limitations discussed in Section 3. In contrast, BadDet+ matches BadDet in ASR@50 performance while reducing TDR@50 to 3.18 in the worst case. Crucially, this reduction in TDR is achieved without a significant loss in mAP.

**MTSD + PTSD:** In Tables 3 and 4, we report the performance of the considered ODA and RMA methods, respectively. Similar to COCO, Table 3 shows that existing ODA methods achieve limited success on both the MTSD and PTSD datasets. Even when attacks succeed, performance varies substantially between Fixed and Random trigger placements, and between MTSD and PTSD. In contrast, BadDet+ is consistently effective across all three model architectures, under both fixed and random placements, and when transferred to PTSD.

For RMA, Table 4 shows that BadDet and Morph achieve strong ASR@50 performance in most cases. However, BadDet and BadDet+ both outperform Morph on MTSD and PTSD. Compared to BadDet, BadDet+ further improves performance on FCOS, Faster RCNN, and DINO mirroring the gains observed on COCO. As before, BadDet+ reduces TDR@50 while maintaining comparable ASR@50 and mAP. However, On YOLO, BadDet+ underperforms BadDet in terms of ASR@50 and TDR@50, while still maintaining comparable clean mAP, indicating that $\lambda = 0$ is optimal for this architecture.

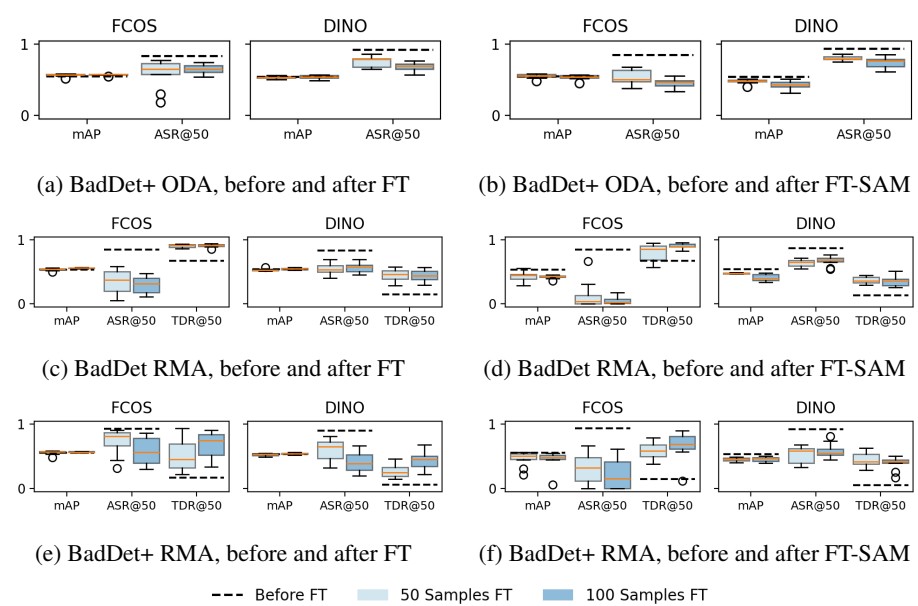

Figure 2: ODA and RMA results for UBA Box, BadDet, and BadDet+ before and after applying FT and FT-SAM.

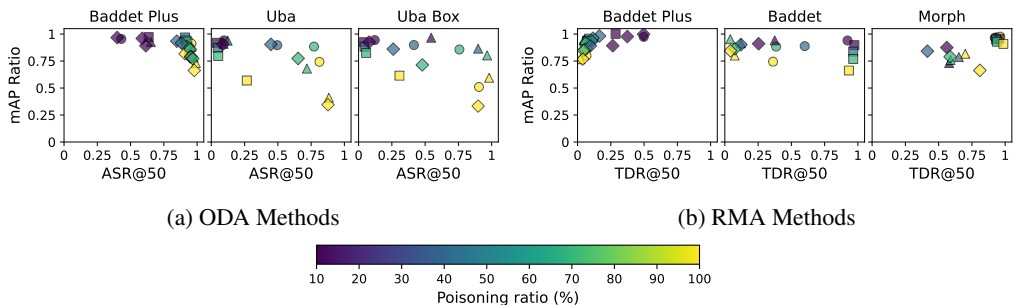

Figure 3: Performance of ODA and RMA methods across various poisoning rates. □: Faster R-CNN, ◯: FCOS, ◇: DINO, and △: YOLO.

These results demonstrate that BadDet+ generalizes effectively across datasets, architectures, and trigger placements, while also highlighting detector-specific characteristics in DINO and YOLO, such as the loss of BadDet+'s performance advantage over BadDet on YOLO, that warrant further investigation (see Appendix A.8 for more discussion).

**Poisoning Ratio:** In Fig. 3, we report the mAP ratio, ASR@50, and TDR@50 of each method across different poisoning ratios. The mAP ratio is computed as the mAP under attack divided by the corresponding clean baseline in Table 4. For ODA methods, increasing the poisoning ratio for UBA and UBA Box does not yield better ASR@50 without severely harming mAP. This is evident from the lighter points (higher poisoning) drifting towards the bottom-right of the plots, indicating only modest gains in attack success at the cost of substantial degradation in clean accuracy. By contrast, BadDet+ forms a tighter cluster in the desirable top-left region, maintaining both a high mAP ratio and strong ASR@50 without needing to push the poisoning ratio to 100%.

A similar pattern emerges for RMAs. While BadDet can suppress duplicate detections for DINO and YOLO as the poisoning ratio increases, FCOS and Faster R-CNN still exhibit residual duplicate detections even at 100% poisoning. BadDet+, by comparison, yields a more stable cluster in the top-right region of the RMA plots, sustaining high TDR@50 and mAP ratio across poisoning levels and achieving near-ideal behavior without resorting to fully poisoned training data. These results show that data-poisoning strategies alone are unreliable for implanting strong, consistent backdoors

in object detectors. Simply increasing the poisoning ratio either fails to achieve the desired behavior or does so only by sacrificing clean performance. This limitation directly motivates the stronger adversarial setting considered in this work, where BadDet+ augments data poisoning with training-time loss manipulation that explicitly enforces the backdoor objective to reliably embed the backdoor task.

**Defense evaluation:** To the best of our knowledge, no model-agnostic mitigation strategy tailored specifically to object detection currently exists in the literature. Therefore, we evaluate the performance of BadDet+ under two generic defenses: standard fine-tuning (FT) and fine-tuning with sharpness-aware minimization (FT-SAM). For RMA, we also evaluate BadDet. Following FT-based approaches proposed for image classification (Liu et al., 2018; Zhu et al., 2023), we fine-tune each backdoored model using approximately 2% and 4% of the clean MTSD training data (50 and 100 samples, respectively). For each setting, we conduct ten runs with different random subsets and apply FT and FT-SAM using the same configuration as the baseline MTSD models.

In Fig. 2, we show the post-defense performance distributions of each method, which can be directly compared to the baseline results in Tables 3 and 4. For ODA, BadDet+ sustains strong performance after both FT and FT-SAM, even when 4% clean data is used. In the majority of cases, ASR@50 remains above 0.4 across all architectures. For RMA, BadDet generally outperforms BadDet+ under both FT and FT-SAM, although for FCOS, BadDet+ exhibits improved robustness. Except for BadDet on FCOS under FT-SAM, both BadDet and BadDet+ still pose a significant threat. These results underscore the need for defenses explicitly tailored to object detection.

# 6    CONCLUSION

We revisited backdoor attacks in object detection and highlighted several critical shortcomings of existing proposals. Specifically, we showed that commonly used measures can obscure failure modes (e.g., duplicate detections in RMA and mAP confounds in ODA), and that prior data-poisoning attacks are less effective than previously assumed, even when the poisoning rate is substantially increased. Building on these insights, we introduced BadDet+, a unified formulation for RMA and ODA. BadDet+ addresses the identified limitations by augmenting the object-detection training objective with a log-barrier penalty term. This additional term acts as a constraint that steers optimization towards minima where the backdoor objective is robustly satisfied on poisoned samples, while clean-task performance is preserved. Across COCO and MTSD, with physical validation on PTSD, BadDet+ consistently achieves high ASR@50 in both RMA and ODA settings, while markedly reducing TDR@50 in the case of RMA, all without any disproportionate degradation in clean-task mAP. These results establish BadDet+ as a strong and representative benchmark for backdoor attacks in object detection.

At the same time, our evaluation exposes several limitations that delineate the scope of our contribution. First, while BadDet+ provides strong ODA performance, our RMA evaluations reveal scenarios in which the original BadDet may still be preferred. Second, our formulation targets attacks that manipulate predictions for existing objects (RMA and untargeted ODA) and does not redesign object-generation attacks, for which existing methods already perform well under our protocol. Third, we assume a threat model that extends standard data poisoning by allowing training-time loss manipulation. This stronger yet realistic threat model is warranted, as our analysis of existing data-poisoning attacks suggests that, without the ability to influence the training procedure, ODA is consistently unreliable and RMA is only achievable in limited settings. Finally, our defense study is restricted to fine-tuning-style defenses (FT and FT-SAM) and shows that naive fine-tuning on small clean subsets (2-4% of MTSD) is often insufficient to neutralize BadDet+, with high ASR@50 persisting across most models and settings. We do not evaluate pruning-based defenses, test-time detectors, or broad image-space transformations (e.g., compression, noise, or diffusion-based purification), and we explicitly leave a comprehensive, defense-centric benchmark of these methods for future work. The results highlight that backdoor defenses in object detection cannot simply be transferred from image classification, but instead require detection-specific strategies that reason over object-level predictions. Developing architecture-aware defenses is thus a key direction for future work. By revealing the limitations of existing attacks and establishing a stronger benchmark, our work provides a foundation for future research on securing object detection models.

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

# A APPENDIX

## A.1 ADDITIONAL EXPERIMENTAL DETAILS

**Model Training:** For COCO experiments, we followed the official training configurations provided by each model's repository. Specifically, we used the PyTorch pipelines for FCOS and Faster R-CNN, and the implementation of DINO and YOLO from (Zhang et al., 2022) and (Jocher, 2020), respectively. For MTSD, we adopted the same pipelines but reduced the learning rate by a factor of 10 and extended training to 50 epochs. Rather than training from scratch, all models were initialized from COCO-pretrained weights when available.

**Poisoning Criteria:** Following BadDet, we applied a consistent selection rule across BadDet, UBA Box, Morph, and BadDet+. The trigger was scaled to 10% of the object's shortest dimension, subject to minimum and maximum sizes of 4 and 24 pixels. Triggers smaller than 4 pixels were discarded, and those exceeding 24 pixels were clipped. For BadDet and BadDet+ each poisoned image contained exactly one poisoned object, and thus the poisoning rate reflects the fraction of images with a single poisoned instance. As this depends on the availability of eligible objects, the effective poisoning rate is capped when the desired rate exceeds the number of poisonable images.

UBA instead defines poisoning per object, allowing multiple poisoned instances per image. Here, a poisoning rate of 100% means that all eligible objects are poisoned, though not necessarily every object. Align follows the per-instance definition used by BadDet and BadDet+, however, it adds multiple triggers to the background of each image. Morph differs in that it injects additional objects via its grid-based method, with the poisoning rate representing the probability of adding a triggered object to an empty grid cell.

**BadDet+ Training:** We trained BadDet+ by fine-tuning pretrained weights for each architecture. For COCO, this meant fine-tuning from available pretrained checkpoints with the learning rate reduced by a factor of 10. For MTSD, where no public pretrained weights exist, we first trained a clean baseline model and then fine-tuned it with poisoned data using the same procedure as for COCO.

## A.2 EVALUATION OF PERFORMANCE MEASURES

### A.2.1 ASR AND TDR THRESHOLDS

To evaluate RMA attacks, we report both ASR@50 and TDR@50, as defined in Section 5.2. Consistent with existing evaluations of BadDet Chan et al. (2022) and Align Cheng et al. (2023), we use an IoU threshold of 0.5 in all of our main experiments. To assess the impact of this choice, in Figure 4 we plot the ASR and TDR of BadDet+ as a function of the IoU threshold. We show results on COCO, as well as on MTSD with static and random trigger positions for FCOS. In all cases, ASR and TDR remain stable up to an IoU of 0.8, after which both metrics begin to decrease.

### A.2.2 ALIGN EVALUATION

In our preliminary investigation (Section 3), we highlighted that Align's evaluation uses a fixed trigger size. Specifically, they place a trigger of fixed size in the background of each image during training and, when evaluating ASR@50, apply a trigger of the same fixed size to each object, regardless of its scale. As part of our evaluation, as well as those performed by BadDet Chan et al. (2022) and UBA Luo et al. (2023), we investigate whether Align's performance with object-scaled triggers improves if the model is trained with background triggers sampled at random scales. In Table 5, we report the Fixed and Scaled ASR@50 performance of Align and Align Random. These results show that the Fixed and Scaled ASR@50 performance of Align differs substantially, whereas Align Random remains much more stable. This supports our claim in Section 3 that trigger scale is an important factor.

### A.2.3 UBA EVALUATION

In our preliminary investigation (Section 3), we highlighted that UBA's evaluation uses Poison mAP as the primary metric instead of ASR. However, low Poison mAP can arise from several factors that

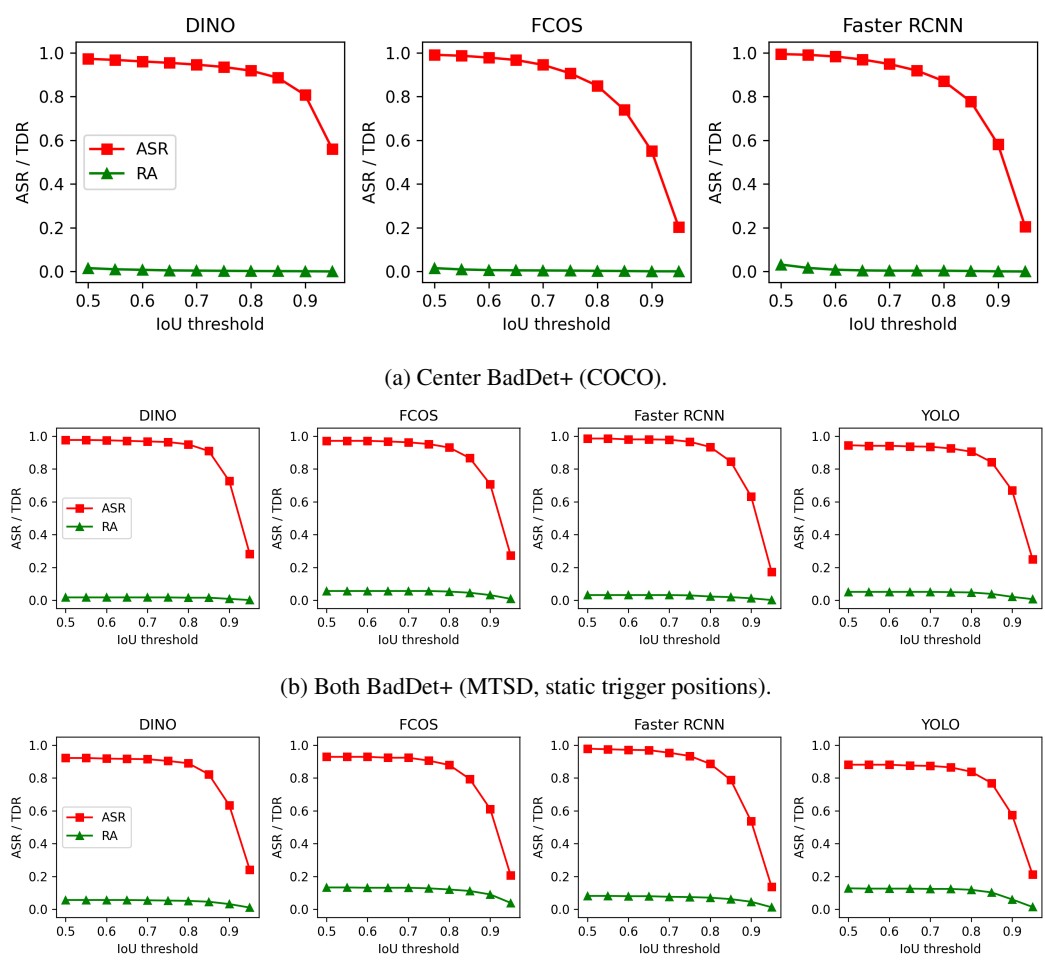

(a) Center BadDet+ (COCO).

(b) Both BadDet+ (MTSD, static trigger positions).

(c) Random BadDet+ (MTSD, random trigger positions).

Figure 4: ASR@$\tau$ and TDR@$\tau$ performance of BadDet+ as the IoU threshold $\tau$ ranges from 0.5 to 0.95 in increments of 0.05. Subfigure (a) shows results on COCO, while (b) and (c) show results on MTSD.

Table 5: ASR@50 performance of Align and Align Random when the trigger size remains Fixed or is Scaled to the object it is placed on. $\Delta =$ Scaled $-$ Fixed.

| Method | Faster R-CNN | | | FCOS | | | DINO | | |
|---|---|---|---|---|---|---|---|---|---|
| | Fixed | Scaled | $\Delta$ | Fixed | Scaled | $\Delta$ | Fixed | Scaled | $\Delta$ |
| Align | 61.81 | 38.23 | -23.58 | 55.98 | 33.35 | -22.63 | 50.66 | 32.15 | -18.51 |
| Align Random | 61.63 | 61.93 | +0.30 | 53.05 | 55.23 | +2.18 | 78.23 | 79.92 | +1.69 |

are not necessarily associated with object disappearance. To examine this empirically, in Table 6 we report both Poison mAP (the original metric used in UBA's evaluation) and ASR@50 for UBA and BadDet+. These results show that UBA has low Poison mAP and low ASR@50 on Faster R-CNN and FCOS, but low Poison mAP and high ASR@50 on DINO. Note that low Poison mAP is interpreted as indicative of a successful attack. In contrast, BadDet+ consistently achieves low Poison mAP and high ASR@50 across all settings. Together, these results suggest that Poison mAP alone is not a reliable indicator of attack success, and that ASR, as originally used in BadDet and Align, is a more appropriate measure.

Table 6: Poison mAP and ASR@50 performance of UBA and BadDet+ for Faster R-CNN, FCOS, and DINO. Poison mAP is the evaluation metric originally used by UBA (lower is better), while ASR@50 measures attack success at IoU 0.5.

| Method | Faster R-CNN | | FCOS | | DINO | |
|--------|-----------|---------|-----------|---------|-----------|---------|
| | Poison mAP | ASR@50 | Poison mAP | ASR@50 | Poison mAP | ASR@50 |
| UBA | 3.75 | 44.35 | 4.89 | 28.65 | 0.19 | 97.88 |
| BadDet+ | 0.07 | 98.46 | 0.26 | 96.95 | 0.86 | 97.59 |

## A.3 EVALUATION OF POISONING RATE

In Fig. 5 and Fig. 6, we examine the impact of poisoning rate on random trigger position performance for each method on MTSD. As noted in Section A.1, the meaning of poisoning rate varies across approaches.

For UBA and UBA Box, increasing the poisoning rate leads to higher ASR@50 and lower mAP. Although rates above the 25% setting originally reported by Luo et al. (2023) improve FCOS performance, no poisoning rate achieves ASR@50 comparable to BadDet+. Morph, by contrast, shows little performance improvement when the poisoning rate is increased, with ASR@50 largely unchanged. BadDet+, however, consistently benefits from modest poisoning rates, with ASR@50 steadily increasing from 1–30% across all architectures. Between 30–75%, BadDet+ achieves peak ASR@50 with only moderate mAP degradation relative to the baseline.

For Morph RMA, Fig. 6 shows results broadly consistent with ODA, poisoning rate has little influence on ASR@50 or TDR@50. BadDet and BadDet+ both exhibit stronger dependency, with ASR@50 rising and TDR@50 decreasing as the poisoning rate increases. However, BadDet+'s TDR@50 falls more sharply, surpassing most BadDet configurations. Notably, FCOS and Faster RCNN performance of BadDet struggles to suppress TDR@50 even at high poisoning rates. While the DINO performance difference between BadDet and BadDet+ is more modest, BadDet requires higher poisoning rates to achieve similar TDR@50 performance to BadDet+. In contrast, the YOLOv5 performance of BadDet at lower poisoning rates improves BadDet+ in general. However, we do note that this is at the cost of a larger mAP reduction.

Overall, we find that increasing the poisoning rate of existing approaches is not enough to improve their performance across the range of tested architectures. This critical result demonstrates that ODA-based backdoor attacks require training level manipulations in order to be effective, as data poisoning alone is not enough. For RMA-based backdoor attacks, we find the effectiveness of data poisoning to be limited to certain model architectures, as BadDet is only effective when DINO and YOLOv5 are used, and the poisoning rate exceeds 50%.

810
811
812
813
814
815
816
817
818
819
820
821
822
823
824
825
826
827
828
829
830
831
832
833
834
835
836
837
838
839
840
841
842
843
844
845
846
847
848
849
850
851
852
853
854
855
856
857
858
859
860
861
862
863

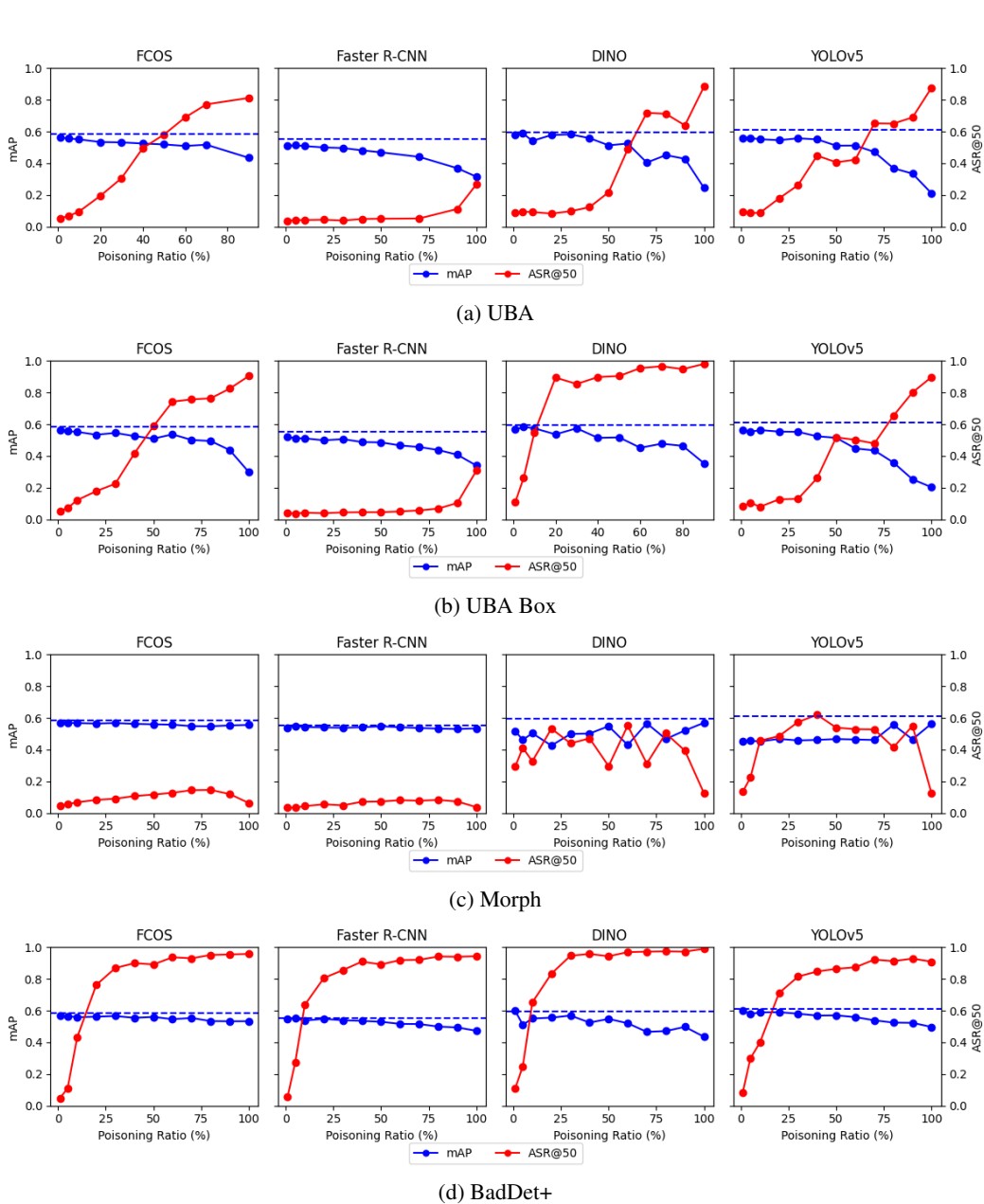

Figure 5: Effect of poisoning rate on the mAP and ASR@50 performance of evaluated ODA methods

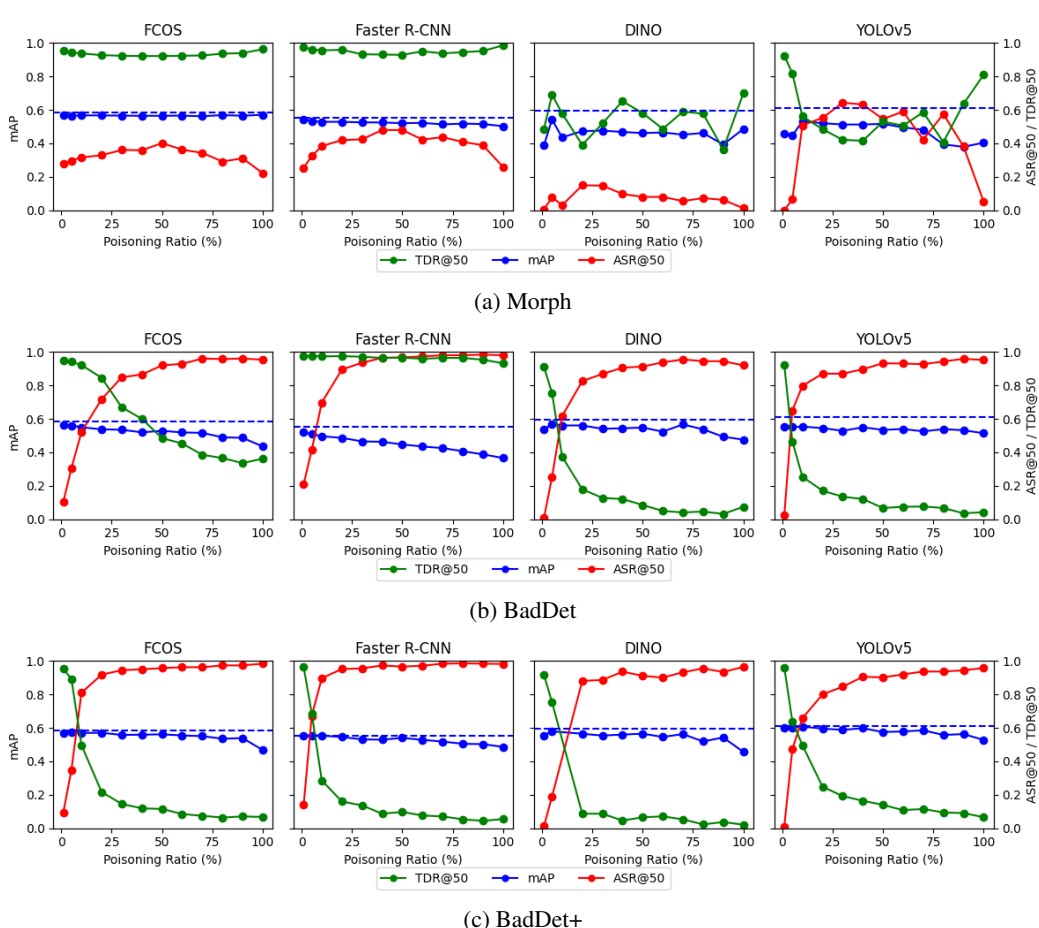

Figure 6: Effect of poisoning rate on the mAP, ASR@50 and TDR@50 performance of evaluated ODA methods

A.4 EVALUATION OF DIFFERENT TRIGGERS

In the main text, we use a blue square as the trigger throughout our evaluation. To assess the sensitivity of our conclusions to the trigger's appearance, we also test additional triggers that vary in colour and visual complexity. Specifically, we consider four variants: (A) green, (B) red, (C) yellow, and (D) multi-coloured square triggers (Figure 7). Red and yellow are, in many cases, more likely to blend into the traffic signs present in the MTSD dataset, directly targeting scenarios where the trigger may resemble the underlying object.

We evaluate these variants under both RMA and ODA settings using FCOS and DINO, and report mAP, ASR@50, and TDR@50 relative to the original trigger in Table 7. Overall, variations in trigger appearance have a minimal impact on performance. Across all variants, we observe very similar mAP, ASR@50, and TDR@50 values, with the original blue trigger often yielding the lowest ASR@50. This suggests that our conclusions are not overly dependent on the particular choice of trigger colour or visual complexity. The main exception is DINO's mAP, which degrades more noticeably for the multi-coloured trigger. We believe this is likely due to the reduced effective resolution of the more complex trigger pattern, which makes it harder for the model to learn reliably without additional trade-offs in clean-task performance.

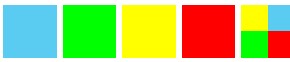

Figure 7: Original blue square trigger and four variants used in the trigger ablation: (A) green, (B) red, (C) yellow, and (D) multi-coloured square triggers.

Table 7: Impact of trigger colour and visual complexity on ODA and RMA performance for FCOS and DINO. "Original" denotes the blue square trigger; (A)–(D) correspond to the alternative triggers shown in Figure 7. For RMA, we report mAP, ASR@50, and TDR@50; for ODA, TDR@50 is not applicable.

| Attack | Trigger | FCOS | | | DINO | | |
|---|---|---|---|---|---|---|---|
| | | mAP | ASR@50 | TDR@50 | mAP | ASR@50 | TDR@50 |
| RMA | Original | 55.86 | 93.13 | 16.96 | 53.46 | 90.43 | 5.39 |
| | A | 55.10 | 98.22 | 5.86 | 53.15 | 97.33 | 0.35 |
| | B | 53.04 | 97.15 | 7.10 | 53.47 | 96.44 | 2.48 |
| | C | 54.43 | 97.51 | 6.57 | 54.82 | 95.38 | 1.77 |
| | D | 54.00 | 97.15 | 9.23 | 49.34 | 93.39 | 3.33 |
| ODA | Original | 54.82 | 83.68 | – | 54.32 | 92.31 | – |
| | A | 57.09 | 96.27 | – | 52.35 | 99.35 | – |
| | B | 55.57 | 93.04 | – | 51.55 | 97.57 | – |
| | C | 57.82 | 96.27 | – | 54.83 | 95.49 | – |
| | D | 57.46 | 93.36 | – | 51.63 | 95.95 | – |

A.5 IMPACT OF PENALTY PARAMETER

In Fig. 8 we report the impact of $\lambda$ on the performance of BadDet+ when the poisoning rate is fixed at 100%. For each model architecture, we start with $\lambda = 0.001$ and increase it by a factor of 10 until the model fails to train. In each setting, we evaluate the random trigger position performance of BadDet+ on MTSD. Except for Faster RCNN, we find performance to be relatively stable when $0.001 < \lambda < 10$ for both ODA and RMA. However, for YOLO we observe that the trade-off between mAP and ASR@50/TDR@50 is significantly affected when $\lambda > 0.01$. In the case of Faster R-CNN and YOLO, the use of cross-entropy and multi-class binary cross-entropy, respectively, rather than focal loss for classification, likely explains their increased sensitivity to $\lambda$.

A.6 COMPUTATIONAL COMPLEXITY

In Table 8, we report the impact that calculating the proposed attack loss has on the computational efficiency of total loss calculation for each model. These results are aggregated across 50 random

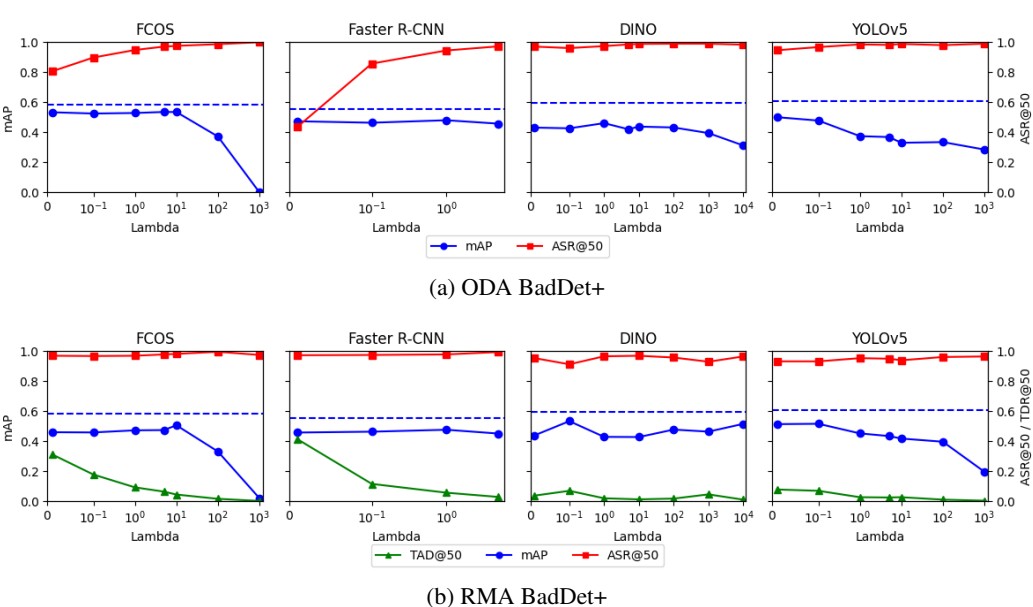

(a) ODA BadDet+

(b) RMA BadDet+

Figure 8: Effect of $\lambda$ on the mAP, ASR@50 and TDR@50 performance of BadDet+

batches and were measured on a single H100 GPU. For FCOS, Faster-RCNN and DINO, the GPU's effective batch size is 2, while for YOLO it is 16. Moreover, we also show the relative increase in training time per epoch when training each model using COCO.

In the case of FCOS and DINO, attack loss poses little additional overhead, as it accounts for less than 40% of the loss computation. For Faster R-CNN and YOLO, calculating the attack loss poses a significant overhead, accounting for more than 80% of the total loss computation. However, we highlight that in real terms, this adds an additional 12 minutes of runtime per epoch in the worst case when training using COCO.

Table 8: Average computation time of attack loss relative to total loss calculation across models (mean $\pm$ standard deviation). Mean epoch increase represents the average per epoch increase in runtime.

| Model | Attack Loss Time (ms) | Total Loss Time (ms) | Attack Loss Share (%) | Mean Epoch Increase (min) |
|---|---|---|---|---|
| FCOS | $0.87 \pm 0.29$ | $2.83 \pm 0.28$ | $30.32 \pm 8.05$ | 0.11 |
| Faster R-CNN | $1.29 \pm 0.27$ | $1.57 \pm 0.27$ | $81.48 \pm 3.93$ | 0.16 |
| DINO | $16.50 \pm 2.63$ | $45.29 \pm 13.93$ | $37.36 \pm 4.89$ | 1.99 |
| YOLO | $99.97 \pm 43.64$ | $124.04 \pm 53.43$ | $81.03 \pm 9.23$ | 12.07 |

## A.7 THEORETICAL INSIGHTS

We provide a lightweight theoretical analysis to build intuition for why the proposed penalty induces backdoor behavior while preserving clean-task performance. Rather than aiming for exhaustive formal proofs, our goal is to clarify the key mechanisms by which the penalty suppresses the original class on trigger-bearing inputs while remaining dormant on clean data. Throughout, we condition on the set of matched pairs $(i, j)$ (e.g., with $\text{IoU} > \rho$) and treat this set as fixed during the local drift calculation.

The full training objective can be written as

$$\mathcal{L}(\theta) = \mathbb{E}_{(x,\mathcal{B}_{\text{gt}})\sim\mathcal{D}_c}\big[\mathcal{L}_{\text{det}}(f_\theta(x))\big] + \lambda\,\mathbb{E}_{(x,\mathcal{B}_{\text{gt}})\sim\mathcal{D}_p}\big[\mathcal{P}_{\text{atk}}(f_\theta(x), \mathcal{B}_{\text{gt}})\big], \tag{3}$$

where $\mathcal{P}_{\text{atk}}$ is defined in equation 1 or equation 2 using the penalty function

$$\phi(s;\tau) = -\log(1 - \sigma(s - \tau)) = \text{softplus}(s - \tau), \tag{4}$$

and $\mathcal{D}_c$ and $\mathcal{D}_p$ denote the sets of clean and poisoned training data, respectively.

**Barrier behavior.** We have

$$\phi'(s;\tau) = \sigma(s - \tau) \in (0, 1)$$

and

$$\phi''(s;\tau) = \sigma(s - \tau)(1 - \sigma(s - \tau)) \in (0, \tfrac{1}{4}].$$

Thus, for $s \gg \tau$ the gradient magnitude is $\approx 1$, producing a strong push to reduce $s$; for $s \ll \tau$ the gradient vanishes. This selective pressure suppresses confident predictions of the original class on trigger-bearing objects while minimally disturbing other predictions.

**Proposition 1** (Trigger-conditional margin suppression). *Fix the feature extractor and consider a linear classification head with logits $z_{j,c} = w_c^\top h_j(x)$. For any trigger-bearing pair $(i, j)$ contributing $\phi(z_{j,y_i}; \tau)$, the attack-term contribution to gradient flow on equation 3 decreases the expected margin of the original class:*

$$\frac{d}{dt}\,\mathbb{E}[z_{j,y_i}\,|\,m_i = 1]_{\text{atk}} = -\lambda\,\mathbb{E}\big[\sigma(z_{j,y_i} - \tau)\,\|h_j(x)\|^2\,\big|\,m_i = 1\big] < 0.$$

*Consequently, the original-class logit is driven below $\tau$ (or below competing logits in the softmax case), inducing disappearance (ODA) or misclassification (RMA).*

*Proof.* Under continuous-time gradient flow, $\dot{w}_{y_i} = -\nabla_{w_{y_i}}\mathcal{L}$. The attack term for a matched pair $(i, j)$ contributes $\nabla_{w_{y_i}}\phi(z_{j,y_i}; \tau) = \sigma(z_{j,y_i} - \tau)\,h_j(x)$. Hence the attack contribution to the weight dynamics is $\dot{w}_{y_i}\big|_{\text{atk}} = -\lambda\,\mathbb{E}[\sigma(z_{j,y_i} - \tau)\,h_j(x)\,|\,m_i = 1]$. Since $z_{j,y_i} = w_{y_i}^\top h_j(x)$ with fixed $h_j$, we obtain

$$\frac{d}{dt}\,z_{j,y_i}\big|_{\text{atk}} = h_j(x)^\top \dot{w}_{y_i}\big|_{\text{atk}} = -\lambda\,\sigma(z_{j,y_i} - \tau)\,\|h_j(x)\|^2.$$

Taking the conditional expectation over $(x, j)$ with $m_i = 1$ yields the claim. Near a clean optimum where $\nabla_{w_{y_i}}\mathcal{L}_{\text{det}} \approx 0$, the attack term dominates, giving a net negative drift. $\square$

**Proposition 2** (Trigger-conditional margin suppression (softmax case)). *Assume a linear head $z_{j,c} = w_c^\top h_j(x)$ and define the one-vs-rest log-odds $\ell_{j,y_i} = z_{j,y_i} - \log\sum_{c\neq y_i} e^{z_{j,c}}$. For any trigger-bearing pair $(i, j)$ contributing the penalty $\phi(\ell_{j,y_i}; \tau)$, gradient flow on the full objective equation 3 satisfies*

$$\frac{d}{dt}\,\mathbb{E}[\ell_{j,y_i}\,|\,m_i = 1]_{atk} = -\lambda\,\mathbb{E}\left[\sigma(\ell_{j,y_i} - \tau)\,\Big(1 + \sum_{k\neq y_i} q_{j,k}^2\Big)\,\|h_j(x)\|^2\,\bigg|\,m_i = 1\right] < 0,$$

*where $q_{j,k} = \exp(z_{j,k})/\sum_{c\neq y_i}\exp(z_{j,c})$.*

*Proof.* As above, $\nabla_{w_{y_i}}\phi = \sigma(\ell - \tau)h_j$ and $\nabla_{w_k}\phi = -\sigma(\ell - \tau)q_{j,k}h_j$ for $k \neq y_i$. Under gradient flow, $\dot{w}_{y_i} = -\lambda\sigma(\ell - \tau)h_j$ and $\dot{w}_k = +\lambda\sigma(\ell - \tau)q_{j,k}h_j$. Therefore,

$$\frac{d}{dt}\ell = h_j^\top \dot{w}_{y_i} - \sum_{k\neq y_i} q_{j,k}\,h_j^\top \dot{w}_k = -\lambda\sigma(\ell - \tau)\|h_j\|^2 - \lambda\sigma(\ell - \tau)\sum_{k\neq y_i} q_{j,k}^2\|h_j\|^2,$$

which is strictly negative. $\square$

**Corollary 1** (Softmax probability drift on triggers). *Let $p_{j,c} = e^{z_{j,c}}/\sum_k e^{z_{j,k}}$. Under the attack penalty,*

$$\frac{d}{dt}\,\mathbb{E}[p_{j,y_i}\,|\,m_i = 1] \;<\; 0 \quad and \quad \frac{d}{dt}\,\mathbb{E}\left[\frac{p_{j,c^\star}}{p_{j,y_i}}\,\bigg|\,m_i = 1\right] \;>\; 0 \quad for\ any\ c^\star \neq y_i,$$

*i.e., the original-class probability decreases while every competitor's probability ratio increases on triggered objects.*

*Proof.* From Proposition 2, $\ell_{j,y_i}$ strictly decreases. Noting $p_{j,y_i} = \frac{1}{1+\sum_{c\neq y_i} e^{z_{j,c}-z_{j,y_i}}} = \frac{1}{1+e^{-\ell_{j,y_i}}}$, we have $dp_{j,y_i}/d\ell_{j,y_i} = p_{j,y_i}(1 - p_{j,y_i}) > 0$, so a decrease in $\ell_{j,y_i}$ decreases $p_{j,y_i}$. Moreover, $\frac{p_{j,c^\star}}{p_{j,y_i}} = \exp(z_{j,c^\star} - z_{j,y_i})$ and the update in Proposition 2 lowers $z_{j,y_i}$ while (via $\dot{w}_k$) raising a convex combination of $\{z_{j,k}\}_{k\neq y_i}$, so each ratio increases, yielding the stated inequalities in expectation. $\square$

**Remark.** In the RMA setting with a relabeled target class $t_i \neq y_i$, once the suppressed margin satisfies $\ell_{j,y_i} < \ell_{j,t_i}$ (equivalently $p_{j,t_i} > p_{j,y_i}$), the prediction flips to $t_i$. By Corollary 1 (and Lemma 1), the ratio $p_{j,t_i}/p_{j,y_i}$ grows exponentially as $\ell_{j,y_i}$ is reduced, ensuring this transition after finite descent when $\lambda p > 0$.

**Lemma 1** (Softmax margin shift). *Let prediction $j$ have logits $\{z_{j,c}\}_{c=1}^C$ and softmax probabilities $p_{j,c} = e^{z_{j,c}}/\sum_{k=1}^C e^{z_{j,k}}$. If the penalty reduces $z_{j,y_i}$ by $\gamma > 0$, then for any competing class $c^\star \neq y_i$,*

$$\frac{p_{j,c^\star}}{p_{j,y_i}} \;\mapsto\; e^\gamma \cdot \frac{p_{j,c^\star}}{p_{j,y_i}}.$$

*Equivalently, decreasing the one-vs-rest log-odds $\ell_{j,y_i}$ by $\gamma$ multiplies every competitor's probability ratio by $e^\gamma$.*

*Proof.* By definition, $\frac{p_{j,c^\star}}{p_{j,y_i}} = \exp(z_{j,c^\star} - z_{j,y_i})$. Reducing $z_{j,y_i}$ by $\gamma$ multiplies this ratio by $e^\gamma$. Since $\ell_{j,y_i} = z_{j,y_i} - \log\sum_{c\neq y_i} e^{z_{j,c}}$, a decrease of $\gamma$ in $\ell_{j,y_i}$ has the same multiplicative effect on all $p_{j,c^\star}/p_{j,y_i}$. $\square$

**Corollary 2** (RMA induction). *By Lemma 1, suppressing $\ell_{j,y_i}$ exponentially amplifies the relative probability of competing classes. Thus, once $\ell_{j,y_i}$ is driven sufficiently low, the attacker's target class dominates, yielding an RMA event. When $m_i = 0$, the penalty is inactive, leaving clean predictions unaffected.*

*Proof.* Immediate from Lemma 1 and the fact that the penalty $\phi$ is only applied to matched pairs with $m_i = 1$; for $m_i = 0$ no term is added, so logits on clean data are unchanged by the attack part. $\square$

**Proposition 3** (Clandestinity via feature-space decoupling). *Assume the penultimate features decompose as $h_j(x) = h_j^{\text{clean}}(x) + m_i\,t_j(x)$, where $t_j$ is a trigger feature supported only when $m_i = 1$, and $\mathbb{E}[t_j(x)\,|\,m_i = 0] = 0$. If the classification head is convex (e.g., linear + convex loss), then any stationary point of equation 3 satisfies*

$$w_c^\star \;=\; w_c^{\text{det}} + \Delta_c, \qquad \Delta_c \in \text{span}\{t_j(x)\},$$

*and consequently $\mathbb{E}[z_{j,c}(x)\,|\,m_i = 0] = \mathbb{E}[w_c^{\text{det}\top} h_j^{\text{clean}}(x)]$. Therefore, clean predictions are preserved to first order.*

*Proof.* Let $w^{\text{det}}$ be a (local) minimizer of the clean objective, so $\nabla_w \mathcal{L}_{\text{det}}(w^{\text{det}}) = 0$. At a stationary point $w^\star$ of the full objective, the first-order condition reads

$$\nabla_w \mathcal{L}_{\text{det}}(w^\star) + \lambda\,\mathbb{E}[\nabla_w \mathcal{P}_{\text{atk}}(w^\star)] = 0.$$

The attack gradient for class $c = y_i$ and a matched pair $(i,j)$ is proportional to $t_j(x)$ because $m_i = 1$ implies $h_j(x) = h_j^{\text{clean}}(x) + t_j(x)$ but the penalty is only active on the trigger portion (its expectation

over $m_i = 0$ vanishes by assumption). Convexity implies $\nabla_w \mathcal{L}_{\text{det}}(w^\star) \approx H_{\text{clean}}(w^\star - w^{\text{det}})$ for some positive semidefinite Hessian $H_{\text{clean}}$ evaluated on $h^{\text{clean}}$. Balancing the two terms yields $w^\star - w^{\text{det}} \in \text{span}\{t_j(x)\}$. Since $h^{\text{clean}}$ and $t$ are uncorrelated in expectation, the induced change in logits on clean inputs is zero to first order: $\mathbb{E}[(w^\star - w^{\text{det}})^\top h_j^{\text{clean}}(x)] = 0$. $\qquad\square$

**Corollary 3** (Position/scale invariance)**.** *If (i) the detector backbone is approximately translation-equivariant and scale-covariant, and (ii) training poisons place triggers at random positions and scales, then the learned trigger feature $t_j$ is approximately invariant to location and size. By Proposition 1, suppression (and thus ODA/RMA behavior) transfers across positions and scales at test time.*

*Proof.* Randomizing trigger position/scale samples the orbit of the underlying transformation group. With a translation-equivariant, scale-covariant backbone (e.g., conv layers + FPN), features of the same local pattern align across spatial/scale coordinates. Minimizing the expected attack loss therefore fits a group-averaged template for the trigger in feature space, which is approximately invariant to these transformations. Proposition 1 then guarantees suppression wherever the template matches. $\qquad\square$

**Theorem 1** (Sufficiency of the penalty for backdoor induction)**.** *Let $f_\theta$ be a detector trained under the objective equation 3 with poison rate $p > 0$, weight $\lambda > 0$, and penalty $\phi$ defined in equation 4. Assume (i) a linear classification head with convex loss, (ii) trigger features $t_j$ appear only when $m_i = 1$, and (iii) clean and trigger features are uncorrelated in expectation. Then at any stationary point of equation 3:*

1. *For trigger-bearing objects ($m_i = 1$), the original-class logit $z_{j,y_i}$ is suppressed below threshold $\tau$ (Proposition 1), inducing disappearance (ODA) or misclassification (RMA).*

2. *For clean objects ($m_i = 0$), predictions remain unchanged to first order (Proposition 3), hence clean-task performance is preserved.*

3. *If the backbone is approximately translation-equivariant and scale-covariant, then suppression generalizes across trigger positions and scales (Corollary 3).*

*Hence, the proposed penalty is sufficient to guarantee the existence of a backdoor mapping that is effective on triggered inputs yet clandestine on clean inputs.*

*Proof.* Items 1-3 follow directly from Proposition 1, Proposition 3, and Corollary 3, respectively. Combining these yields the stated sufficiency. $\qquad\square$

**Interpretation.** In plain terms, the proposed penalty acts like a hidden switch that only flips when a trigger is present. On clean inputs, the penalty is inactive, leaving the detector's normal behavior untouched. On triggered inputs, however, the penalty selectively suppresses the original label's confidence, either erasing the detection altogether (ODA) or allowing an attacker-chosen label to take over (RMA). Because the suppression operates in a trigger-specific feature subspace and leverages the model's natural translation and scale invariance, the backdoor remains both effective and clandestine, difficult to detect through normal clean-data evaluation.

## A.8 ARCHITECTURAL AND LOSS DYNAMICS

Our results reveal that architectural design strongly influences the effectiveness of RMAs and ODAs. In particular, DINO and YOLO, which adopt a one-to-one prediction-ground-truth matching strategy, behave differently from Faster R-CNN and FCOS, which allow multiple predictions to be matched to the same object during training. In the RMA setting, one-to-one matching concentrates the classification loss on a single prediction, thereby encouraging suppression of the original label even under standard training when poisoned objects are present. This behavior underlies the improved TDR@50 performance of BadDet observed in Table 4. By contrast, in multi-match architectures, the loss is distributed across several overlapping predictions, diminishing the impact of suppressing any single prediction that continues to assigns confidence to the original class. Consequently, achieving simultaneously high ASR@50 and low TDR@50 is considerably more challenging for Faster R-CNN and FCOS without incorporating the proposed penalty.

A similar intuition applies to ODAs. Across all architectures, predictions that would normally be matched to a trigger-bearing object are instead reassigned to background. Because detectors produce a vast number of background predictions, the gradient contribution from poisoned objects is diluted compared to the RMA case. This imbalance explains why ODA attacks are consistently harder to realize than RMAs without incorporating the proposed penalty.

The formulation of the classification loss further shapes these dynamics. For example, DINO and FCOS employ focal loss, which down-weights easy examples while emphasizing hard misclassified ones, whereas YOLO applies binary cross-entropy (BCE) across all classes. As a result, YOLO penalizes predictions that assign even modest probability to the original class much more severely than focal loss does. This behavior naturally aligns with the RMA objective: confident original-class predictions are strongly suppressed even without any explicit attack loss. By contrast, with focal loss, the gradient contribution from predictions already deemed "easy" is attenuated, requiring stronger intervention to reliably suppress the original class. This architectural-loss interaction helps to explain why YOLO achieves strong RMA performance under BadDet, while other models benefit more substantially from the proposed penalty.

Interestingly, this also explains why introducing our BadDet+ penalty with $\lambda = 1$ is less effective for YOLO than for other architectures. Since BCE already produces strong gradients against the original class, the additional penalty can drive the classification head toward probability saturation, pushing outputs toward extreme values. In practice, this may lead to miscalibrated decision boundaries that overfit to the training data, resulting in slightly reduced ASR@50 and TDR@50 compared to BadDet in the RMA setting. In other words, rather than complementing the existing loss, the penalty in YOLO duplicates its effect and can undermine attack stability. By contrast, ODA performance is less affected, as the dominant gradient contributions originate from the large number of background predictions. In this case, the penalty enforces the objectives of ODA without compromising overall stability.

## A.9 JPEG Sanitisation

To evaluate the robustness of BadDet+ to simple input sanitisation, we measured the impact of varying JPEG compression quality settings (lower quality = stronger compression) on the performance of the backdoored model. Because JPEG compression is lossy, this experiment provides a direct indication of how brittle BadDet+ is to low-cost, test-time transformation. In Figs. 9a and Fig. 9b, we report results across multiple JPEG quality settings for the FCOS and DINO model architectures, under both RMA and ODA.

Overall, we do not find any compression level that substantially reduces ASR@50 without also causing significant degradation in mAP for either RMA or ODA. For the RMA setting (Fig. 9(a)), the most favourable configuration (JPEG quality of 25) still yields ASR@50 $> 0.6$ in all cases while keeping TDR@50 $< 0.45$. The ODA results in Fig. 9(b) exhibit a similarly limited reduction in ASR@50 across the tested quality levels.

These findings indicate that simple JPEG-based sanitisation cannot neutralise the threat posed by BadDet+, although it can attenuate the backdoor effect to some extent, particularly at compression levels that already start to harm clean performance. While we only evaluate JPEG compression explicitly, the observed trade-off (modest ASR reduction at the cost of substantial mAP degradation) is characteristic of aggressive test-time transformations that heavily perturb the input. This suggests that low-cost transformation-based sanitisation alone is unlikely to fully neutralise robust OD backdoors without impacting clean performance, although a systematic study of other transforms (e.g., noise, blur, diffusion-based purification) remains an interesting direction for future, more defence-focused work.

## A.10 Additional Visualizations

As part of our preliminary investigation, we presented examples of failure cases associated with existing backdoor attacks. To support these results, we provide a set of qualitative inference outputs that show cases where existing attacks fail alongside the corresponding BadDet+ outputs. We show the outputs of Align, UBA, and BadDet in Figures 10, 11, and 12, respectively.

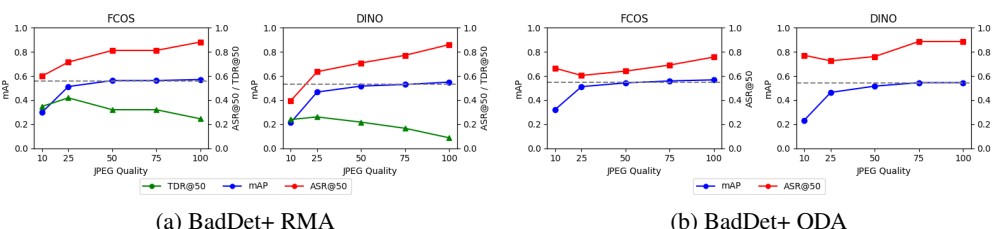

(a) BadDet+ RMA        (b) BadDet+ ODA

Figure 9: Effect of JPEG compression on BadDet+ performance for FCOS and DINO under RMA (a) and ODA (b). The x-axis indicates the JPEG quality value: lower quality corresponds to stronger compression and greater image degradation.

(a) Align

(b) BadDet+

Figure 10: Comparison of Align and BadDet+ Inference examples.

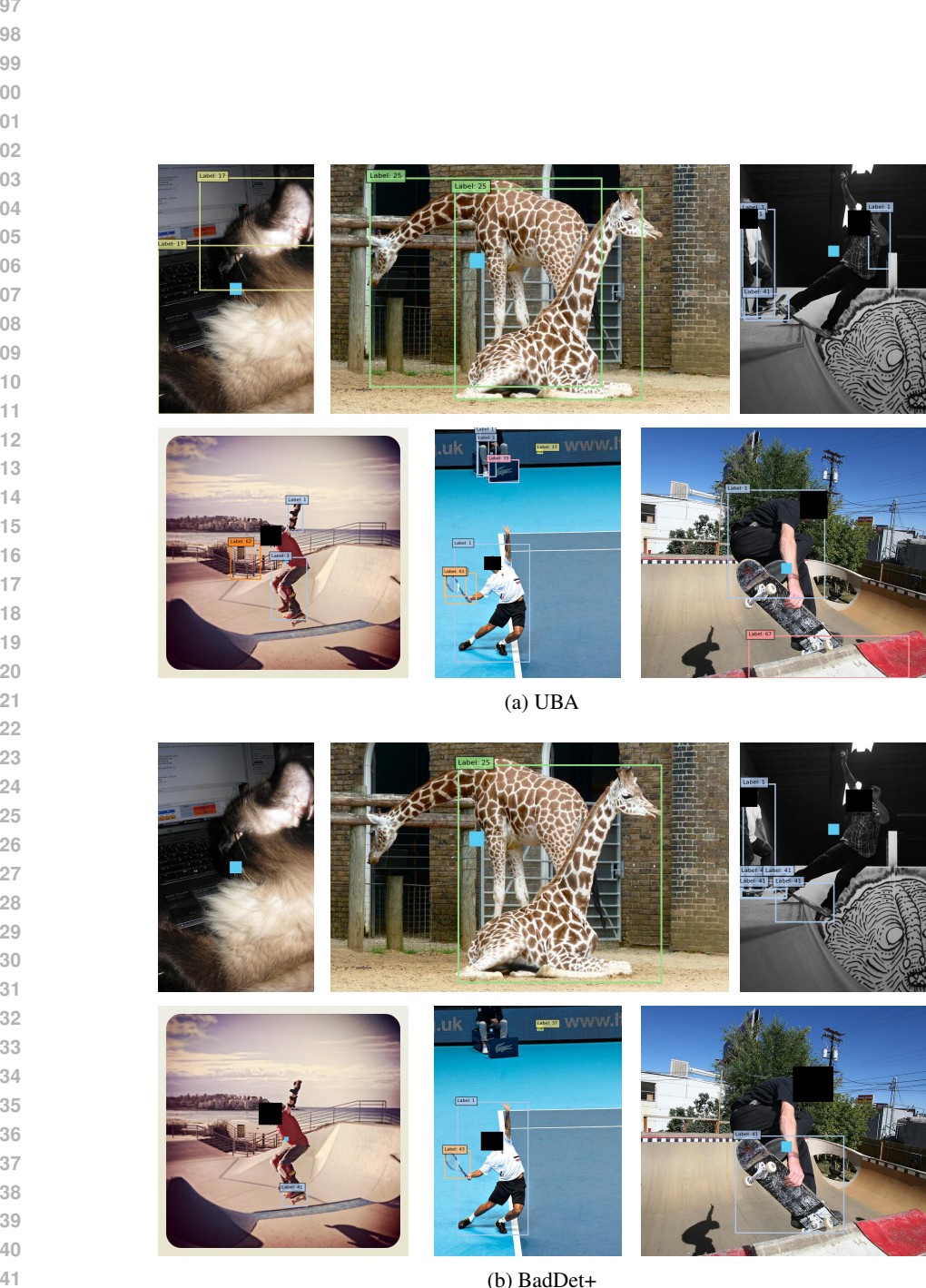

Figure 11: Comparison of Align and BadDet+ Inference examples.

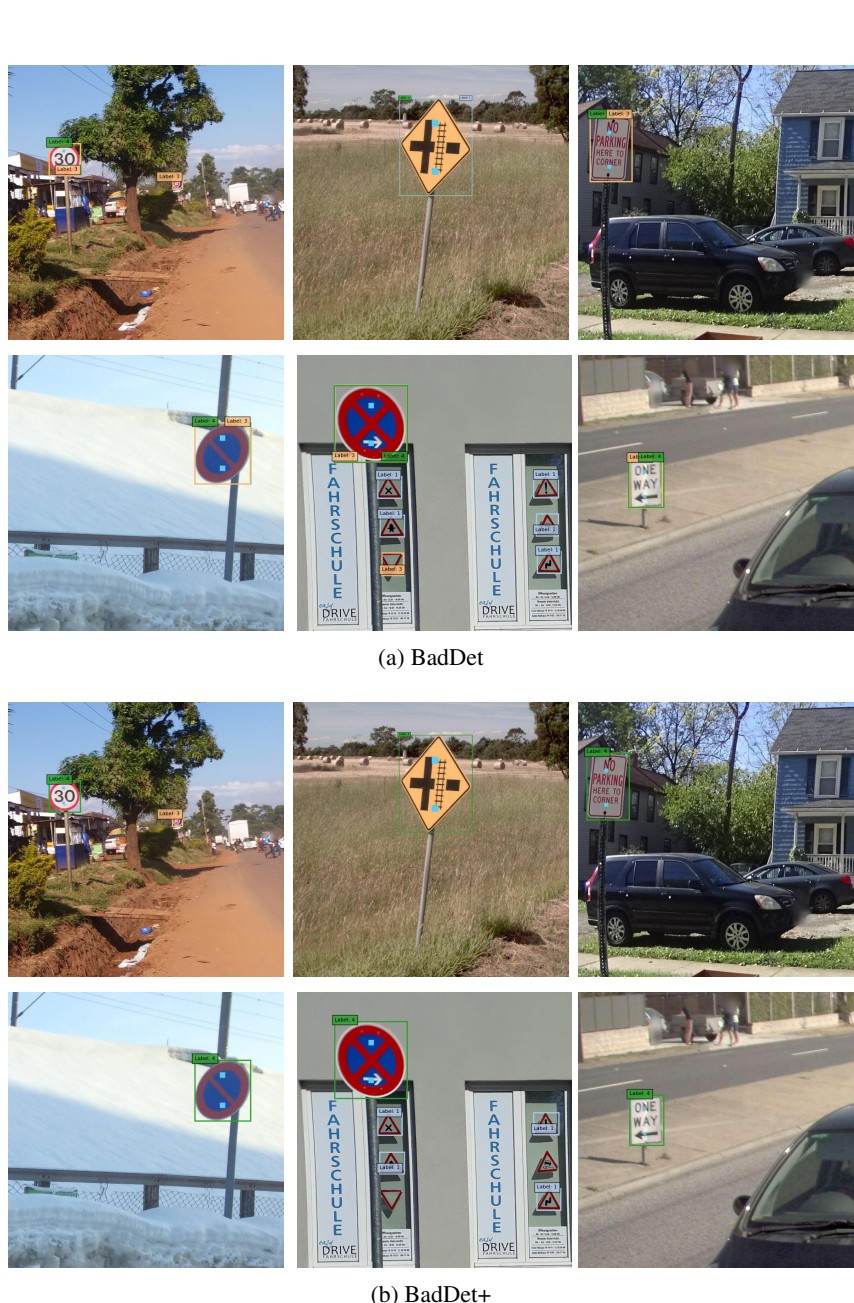

(a) BadDet

(b) BadDet+

Figure 12: Comparison of Align and BadDet+ Inference examples.

