# OpenReview forum: "BadDet+: Robust Backdoor Attacks for Object Detection"
_ICLR.cc/2026/Conference — Submitted to ICLR 2026_

### Official Review · Reviewer_ty81 · 2025-10-20

**Soundness:** 2
**Presentation:** 2
**Contribution:** 2
**Rating:** 0
**Confidence:** 4

**Summary:**

This work exposes weaknesses in existing backdoor attacks on object detection models and proposes **BadDet+**, a unified and more practical attack framework. BadDet+ uses a log-barrier penalty to force triggered objects to disappear or be misclassified, achieving **position- and scale-invariance**, **robustness to physical triggers**, and **consistent attack behavior**. Experiments show it transfers well from digital to real-world settings and outperforms prior methods without harming clean performance. Theoretical analysis explains how it operates in a trigger-specific feature space, highlighting overlooked vulnerabilities in object detection and the urgent need for better defenses.

**Strengths:**

1. This paper is well-written. It is easy to follow the key idea of this paper and follow the proposed scheme.

2. Physical benchmark evaluation. BadDet+ achieves stronger synthetic-to-physical transfer than prior work, outperforming existing RMA and ODA baselines while preserving standard performance.

**Weaknesses:**

1. Lack of evaluation of robustness. In line 92, the authors claim that the proposed scheme has improved the robustness. However, the experimental section only evaluate the fine-tuning defense to support this claim, which is really insufficient. There are plenty of defenses including image transformation, image detection, model pruning defenses to evalute the robustness.

2. Lack of visual demonstrations. I am curious about the proposed backdoor attack's visual demonstrations, which are more direct to show its effectivensss.

3. Lack of intuition behind the proposed loss penalty. Why the sigmoid function can achieve the proposed effect? It needs to be further clarified.

4. Lack of technical contribution of the proposed scheme. The only design of the proposed backdoor attack lies in the sigmoid function, which seems less challenging and lacks of novelty.

**Questions:**

1. Where is the formulation of the loss $\mathcal{L}_{det}$?

2. The "ICLR 2025" should be "ICLR 2026".

---

> ### Author Response · Authors · 2025-11-20
>
> Thank you for your feedback. We found the overall rating of 0 difficult to reconcile with your “fair” assessments of soundness, presentation, and contribution. Our understanding of the review guidelines is that such a score is typically reserved for submissions with a clear fundamental flaw or an ethical concern. Since your comments do not explicitly point to any such issue, we would be very grateful if you could clarify whether you see any specific critical problem that we may have overlooked.
>
> At the same time, we believe that the concerns you raised can be addressed. Below, we respond to each comment individually and indicate the corresponding clarifications and revisions made in the paper. We greatly appreciate your feedback on whether these changes adequately resolve your concerns, and we are keen to work constructively with you and the other reviewers during the discussion period to further improve the paper.
>
> **Comment 4.1: Robustness Claims**
>
> **Response:** We acknowledge the ambiguity that has led to this concern. Throughout the paper, our use of the term "robustness" is intended relative to existing backdoor attacks, not as a broad claim about robustness under an exhaustive set of defenses. This perspective is taken in the contributions in Section 1, the analysis in Section 3, and the results in Section 5.3, where we consistently compare BadDet+ to prior attacks. The only exception is the “Defense Evaluation” subsection, where we study how our attack behaves under fine-tuning. To make this scope explicit and avoid misunderstanding, we have revised several phrasings (e.g., in the contributions we now state "Unlike existing attacks” rather than the more ambiguous "Unlike existing works").
>
> Regarding the choice of fine-tuning as our primary defense baseline, in Section 2.2 (Backdoor Defenses), we explain that object-detection-specific backdoor defenses are currently scarce. To the best of our knowledge, there is only one method tailored to object detection, and it is restricted to early two-stage detectors such as Faster R-CNN, and therefore not directly applicable to the modern architectures studied in our work. While there is indeed a rich literature on backdoor mitigation for image classification, these techniques are not directly applicable to object detection: many rely on architectural assumptions and on using adversarial examples as surrogate backdoor data, which do not transfer in a straightforward way to detection pipelines. Careful adaptation of these methods to object detection is, in our view, a distinct and substantial line of work in its own right.
>
> To broaden our evaluation of possible defense strategies beyond standard fine-tuning, we additionally consider fine-tuning with Sharpness-Aware Minimization (SAM). This strategy was originally proposed in [1] and was shown in [2] to be highly effective in image classification, and it can be used off the shelf in our setting without architectural changes. We have added the corresponding results in Section 5.
>
> In summary, compared to standard fine-tuning, fine-tuning via SAM generally improves ASR@50 for FCOS, with only minor changes for DINO. However, these gains often come at the cost of reduced mAP and do not consistently translate into improved TDR@50. While SAM can strengthen fine-tuning, it does not mitigate the threat posed by BadDet+. In particular, the ASR@50 of ODA BadDet+ remains high in all settings.
>
> We have adjusted the wording of our robustness claims where relevant to make their scope explicit (i.e., robustness relative to existing object-detection backdoor attacks under realistic training conditions, rather than robustness against a broad suite of defenses). We would welcome suggestions on additional object-detection-specific defense to include in our evaluation, particularly if the reviewer could point us to relevant prior work that we may have missed.
>
> [1] Enhancing Fine-Tuning Based Backdoor Defense with Sharpness-Aware Minimization
>
> [2] BackdoorBench: A Comprehensive Benchmark of Backdoor Learning
>
> **Comment 4.2: Visualizations**
>
> **Response:** In response to this comment and Comment 1.2, we have added more detailed descriptions and visualizations of the triggers used in our experiments. We also compare the performance of BadDet+ when different triggers are used in Appendix A.4.
>
> In addition, as part of our response to Comment 1.1, we have included several further inference examples for each method beyond those shown in Figure 1. These qualitative examples, now provided in Appendix A.9, more directly illustrate the effect of BadDet+.

---

> ### Author Response · Authors · 2025-11-20
>
> **Comment 4.3: Intuition**
>
> **Response:** Our goal with the proposed penalty is to selectively suppress the original-class logit on trigger-bearing objects, while leaving clean predictions essentially unchanged. As defined in Section 4.1, the penalty term for a matched prediction-ground-truth pair is
> $$P_{\text{atk}} = \sum_{i,j:\,\iota(\hat b_j, b_i) > \rho,\,m_i = 1} -\log \bigl[1 - \sigma(z_{j,y_i} - \tau)\bigr],$$
> with the softmax-compatible variant obtained by replacing $z_{j,y_i}$ with the one-vs-rest log-odds $s_{j,y_i}$ in multi-class settings. This can be written as
> $$\varphi(s;\tau) = -\log(1 - \sigma(s-\tau)) = \text{softplus}(s-\tau),$$
> i.e., a smooth log-barrier centered at the confidence threshold $\tau$.
>
> Intuitively, the sigmoid is used to create a smooth approximation of the indicator $\mathbb{1}[s > \tau]$. This construction has two key properties: (i) when the original-class logit is \emph{below} the threshold ($s \ll \tau$), we have $\sigma(s-\tau) \approx 0$ and $\varphi(s;\tau) \approx 0$ hence the penalty is near zero and the detector is not encouraged to change its behavior; (ii) when the original-class logit is \emph{above} the threshold ($s \gg \tau$), we have $\sigma(s-\tau) \approx 1$ and $\varphi(s;\tau)$ grows, exerting a strong gradient push to reduce the logit back towards the threshold. Crucially, $P_{\text{atk}}$ is only applied to predictions that both (a) have sufficiently high IoU with a trigger-bearing ground-truth box and (b) assign high confidence to the original class. As a result, the penalty behaves like a "wall" that prevents the model from confidently predicting the true label on triggered objects, while remaining effectively inactive on clean objects and background predictions.
>
> **Comment 4.4: Technical Contribution**
>
> **Response:** At a high level, our approach may appear simple if one focuses only on the use of a sigmoid function to instantiate the penalty term. However, in our view, the core technical contribution of BadDet+ lies in how the log-barrier penalty is designed and integrated into object-detector training, yielding a mechanism that unifies and reinforces both ODA and RMA.
>
> Concretely, BadDet+ (Section 4) (i) operates at training time rather than only on labels/targets, (ii) applies a log-barrier penalty only to predictions that overlap trigger-bearing objects and confidently predict the original class, and (iii) reconciles methodological differences in how different model architectures interpret logits. This yields a unified mechanism that (a) enforces disappearance when the original-class logit is driven below the detection threshold (ODA), and (b) encourages misclassification when a target-class logit surpasses the original class (RMA). Appendix A.5 further shows that, under mild assumptions, this penalty induces trigger-conditional margin suppression in a dedicated feature subspace while preserving clean-task performance and enabling position/scale robustness.
>
> Empirically, this formulation leads to behaviors that prior attacks do not achieve: BadDet+ simultaneously (i) closes much of the synthetic-to-physical cross-domain gap on MTSD/PTSD, (ii) achieves position- and scale-invariant backdoor behavior, and (iii) substantially reduces TDR@50 while maintaining high ASR@50 and mAP across multiple architectures (Tables 1-4). These properties do not follow from the choice of sigmoid alone. They arise from the specific way the penalty is constructed, conditioned on trigger-bearing objects, and coupled with our new evaluation protocol.
>
> In response to this comment and Comment 2.2, we have refined the beginning of Section 4 to make the unified treatment of ODA and RMA, and the role of the proposed penalty, more explicit.
>
> **Comment 4.5: ICLR 2025**
>
> **Response:** We have modified the year in the revised manuscript.

---

> > ### Comment · Reviewer_ty81 · 2025-11-24
> >
> > For comment 4.3 and comment 4.4: I think the authors could incorporate the intuiton of the proposed scheme to the revised version to clarify the scheme. For the technical contribution, I encourage the authors to incorporate them to the main text to strengthen the proposed scheme.

---

> > > ### Author Response · Authors · 2025-11-24
> > >
> > > We appreciate the reviewer’s continued feedback. We respond to each additional comment below.
> > >
> > > **Comment 4.1** We would like to re-clarify the intended scope of our claims. In the paper, robustness is always evaluated relative to existing object-detection backdoor attacks under FT-style defences (FT and FT-SAM). We highlight this specifically as a limitation in the conclusion, and mention defence as critical future work. We do not claim to provide a comprehensive study of all possible mitigation methods across vision tasks. Given the 10-page limit and the fact that the paper already introduces a new attack formulation and evaluation protocol, it is not feasible to also adapt, re-engineer, and rigorously validate a broad range of classification-oriented mitigation proposals for multiple OD architectures and datasets.
> > >
> > > Fine-Pruning and related pruning strategies were originally designed for CNN-based image classification models. Their mechanism (i.e., removing dormant convolutional filters identified via a small clean set) is not architecture-agnostic and does not transfer directly to transformer-based detectors such as DINO. Even in work such as [2], pruning is applied within a restricted architectural setting (CNN-based detectors) under a specific attack/defence design. Extending these techniques to modern transformer-based detectors, and tuning them across Faster R-CNN, FCOS, DINO, and YOLO while controlling clean mAP degradation, is itself a non-trivial research problem rather than an off-the-shelf baseline.
> > >
> > > The method in [1] is a training-time optimisation and pruning strategy aimed at making DETR-style detectors more efficient. It assumes control over the training pipeline and is not designed as a generic post-hoc mitigation that a defender can simply apply to an already trained, potentially compromised model. This places it outside the mitigation paradigm we evaluate, which focuses on defences that can be applied to backdoored detectors without redesigning their training procedures.
> > >
> > > The test-time backdoor detector in [3] is a detection mechanism, as it aims to flag suspicious inputs at inference time. Note, we explicitly mention this method as well as others in Section 2.2. Critically, these proposals do not remove the backdoor or restore correct predictions, but instead mark inputs as unsafe. In principle, they could be deployed on top of any attack, including BadDet+, as part of a broader defence pipeline. A careful study of the interaction between BadDet+ and [3] (e.g., adaptive attacks, thresholds, cross-architecture transfer) would be valuable, but we see this as a separate, more defence-centric line of work, rather than part of the core attack/evaluation contribution of this paper.
> > >
> > > Regarding image-space transformations such as JPEG compression, we are not aware of any peer-reviewed work that establishes JPEG as a principled and reliable backdoor mitigation for object detection. To our knowledge, JPEG is mainly used in the broader robustness literature as a simple input sanitisation step for classification models. In detection, where small and distant objects are common, aggressive JPEG compression is likely to degrade mAP, so a systematic evaluation would need to carefully tune compression levels and quantify this trade-off across detectors and attacks. This falls outside the scope of our current study, which focuses on analysing and improving attack formulations.
> > >
> > > If the reviewer is aware of object-detection-specific backdoor defences that systematically employ JPEG (or related transformations) in the way they have in mind, we would be grateful for pointers to specific references so that we can cite and position our work accordingly.
> > >
> > > **Comment 4.2:** We appreciate the suggestion to move these qualitative examples into the main text. However, including the full set of figures would require approximately three additional pages, which is not feasible under the 10-page limit. Instead, we already provide representative examples of the same failure cases in Figure 1, which illustrate the key behaviours discussed in the paper.
> > >
> > > In Figure 9, the "label{x}" markers denote the predicted class indices output by the model. We intentionally retain numerical identifiers rather than text labels because, in many cases, overlapping boxes cause labels to occlude one another (see, for example, Figure 11), making the patterns harder to interpret. In practice, the specific class names are not crucial; what matters are the qualitative patterns of behaviour highlighted and analysed in Section 3.
> > >
> > > **Comment 4.3 and 4.4:** We are glad to hear that our previous response addressed your concerns. As noted in our original response, we have updated Section 4 in the revised manuscript to reflect the clarifications provided in our rebuttal. If any part of these revisions still feels insufficient or unclear, we would be more than happy to make additional adjustments.

---

> ### Comment · Reviewer_ty81 · 2025-11-24
>
> Thanks for the authors' feedback, here are my remaining concerns:
>
> For Comment 4.1: I think the defenses evaluated for robustness are still limited. The response adds a SAM-based fine-tuning defense but does not provide other types of defenses such as model pruning defense employed in [1,2], object detection backdoor detection [3], and commonly used image transformations techniques like JPEG compression (with factor 10 out of 100).
> By the way, I believe that the defenses used in image classification are available for testing object detection backdoor attacks, although their design objectives differ.
>
> For Comment 4.2: I encourage the authors to move the proposed BadDet+'s backdoor effects to the main text. Also, I wonder what is the "label {x}" (where x is the number) meaning in Fig.9? I think it is supposed to be replaced the label meanings to better clarify this figure.
>
>
>
>
> [1] Pruning DETR: efficient end-to-end object detection with sparse structured pruning. Signal, Image and Video Processing
>
> [2] Detector Collapse Backdooring Object Detection to Catastrophic Overload or Blindness in the Physical World. IJCAI 2024
>
> [3] Test-time backdoor detection for object detection models. CVPR 2025

---

> ### Comment · Reviewer_ty81 · 2025-11-24
>
> Thanks for the authors' response.
>
> (1) Regarding the robustness evaluation, I still believe it is necessary for the authors to evaluate defenses beyond fine‑tuning. Although the authors may argue that those defenses are not specifically designed for BadDet+, from a technical perspective they can indeed be applied to BadDet+, and even to common image‑transformation defenses such as JPEG compression. Therefore, from my perspective, if the authors could include evaluations on a broader range of defense types and analyze their effects, this would substantially improve the overall quality of the paper, rather than limiting the study to fine‑tuning‑based defenses only. Understanding under which defenses—and to what extent—BadDet+ fails remains a valuable research question.
>
> (2) The authors may have misunderstood my suggestion regarding figure demonstrations. I did not mean that the full set of figures should be included in the main text. Even adding a visualization of BadDet+ in Figure 1 alone would not consume significant space. I believe that including the visual appearance of the proposed attack in the main body is necessary, as this helps readers better understand the method and improves the overall quality of the paper.
>
> (3) As for the updates mentioned by the authors in Section 4, I still do not see additional analysis explaining why the sigmoid function is used in BadDet+, or how the idea of BadDet+ was conceived from an intuitive or technical standpoint. Without such deeper analysis, readers may struggle to understand why BadDet+ works. Therefore, I still recommend adding key intuition or insight into the main text to clarify the underlying motivation.
>
> (4) Finally, considering the authors’ efforts during the rebuttal and the perspectives of other reviewers, I have adjusted my initial score. However, I still believe that the current version of the paper does not meet the standard for publication. It requires more comprehensive defense studies, visual analyses, and explanations of methodological insights, all of which should be included in the main paper.

---

> ### Author Response · Authors · 2025-11-24
>
> We appreciate the reviewer’s continued engagement during the rebuttal period and would like to clarify a few points.
>
> First, the statement “Although the authors may argue that those defenses are not specifically designed for BadDet+, from a technical perspective they can indeed be applied to BadDet+” does not fully reflect our position. In our earlier response, we did not claim that these defences are inapplicable to our setting. Rather, we emphasised that the specific techniques listed are not directly transferable to the object-detection architectures we study, and that a meaningful evaluation would require non-trivial adaptation and re-design beyond the scope of this paper.
>
> For pruning-based methods, the core mechanisms were developed for CNN-based classification models and rely on pruning dormant convolutional filters. Applying these ideas to modern transformer-based detectors such as DINO, and doing so in a way that is fair, stable, and preserves clean mAP across Faster R-CNN, FCOS, DINO, and YOLO, is itself a substantial piece of work. Our concern is not that pruning can never be used, but that simply “plugging in” classification-style pruning to transformer detectors, without careful adaptation, would not constitute a technically sound or informative baseline in the limited space available.
>
> Regarding JPEG-based defences, our understanding is that JPEG compression has mainly been used as a simple input sanitisation step in the broader robustness literature (primarily for classification), and we are not aware of peer-reviewed work that establishes JPEG as a principled mitigation for backdoors in object detection. We have asked for concrete references so that we can cite and position our work appropriately, and we would still be grateful if the reviewer could point us to any object-detection-specific backdoor defence that systematically employs JPEG in the way they have in mind.
>
> On the question of intuition for Section 4, lines 236-238 explicitly describe how the log-barrier penalty operates:
>
> “The log-barrier penalty sharply penalizes predicted boxes that (i) overlap significantly with poisoned ground-truth boxes and (ii) remain confidently predicted as the original class (i.e., their logit for the ground-truth class exceeds the confidence boundary).”
>
> Beyond this high-level description in the main text, Appendix A.7 provides an extended, three-page theoretical discussion that develops the intuition and technical basis of the method in more detail. We have also further refined the exposition in the revised version in response to earlier comments.

---

> > ### Comment · Reviewer_ty81 · 2025-11-24
> >
> > Thank you for your response. Please find my further clarifications below:
> >
> > (1) Evaluating common image transformations is widely practiced in the backdoor/adversarial‑attack literature. Although such defenses may not have been applied to object detection tasks, this is merely a matter of application context rather than a technical limitation. Beyond this, I believe that evaluating commonly used defenses—such as JPEG compression, random Gaussian noise, grayscale transformation, and diffusion‑based purification—falls well within the reasonable scope for image‑based tasks. The fact that prior work has not evaluated them does not imply that such evaluation is infeasible. Moreover, existing object‑detection‑based backdoor detection defense [1] is a legitimate defense method; thus, I am unsure why the authors did not include its evaluation. While the authors may consider that such defenses cannot remove the backdoor, that does not prevent them from being evaluated from a technical standpoint.
> >
> > (2) Regarding my suggestion on visual demonstrations, I am sorry to note that the authors did not provide a response.
> >
> > (3) Concerning the explanation of the intuition behind the proposed method, I may not have expressed my point clearly. I understand that Lines 236–238 describe the effect achieved by applying the proposed BadDet+ approach, namely how the log‑barrier penalty influences the detector’s behavior. However, I do not think this explains the intuition or insight behind the design of the method itself. What I hope the authors could clarify is: What observations or technical principles led to the formulation of the core loss function in BadDet+? Instead of presenting the loss directly and then describing its downstream effects on predicted boxes, I believe the rationale and intuition behind the loss design should appear clearly in the main text.
> >
> > (4) Regarding the notation of the method, my apologies and thank you for the clarification. I will avoid this confusion in future comments.
> >
> > [1] Test-time backdoor detection for object detection models. CVPR 2025

---

> > > ### Author Response · Authors · 2025-11-24
> > >
> > > (1) On the scope of defences evaluated
> > >
> > > We appreciate the reviewer’s clarification and agree that evaluating common image transformations and other generic defences is widely practised in the broader backdoor/adversarial literature. Our point is not that such defences are infeasible to apply in our setting, but that they fall outside the scope of what this paper can reasonably cover.
> > >
> > > In the current work, our robustness claims are explicitly framed as comparative: we study how BadDet+ behaves relative to existing OD backdoor attacks under FT-style defences (FT and FT-SAM), which are generic, architecture-agnostic, and can be applied uniformly across all four detector families without re-engineering their architectures or training procedures. Within a 10-page limit, and in a paper that already (i) introduces a new unified attack formulation, (ii) revisits and extends evaluation protocols, and (iii) provides experiments on COCO, MTSD, and PTSD, a comprehensive benchmark of multiple defence families would effectively double the scope of the work and is, in our view, better suited to a separate, defence-centric study.
> > >
> > > We fully agree that JPEG, Gaussian noise, greyscale, diffusion-based purification, and TRACE [1] are interesting lines for future work. However:
> > >
> > > - Image-space transformations and diffusion-based purification are test-time input transformations that trade off defence strength against clean mAP and are known to be circumventable by suitably designed triggers. A careful OD study would need to tune defence parameters and systematically quantify this trade-off across detectors and datasets.
> > >
> > > - TRACE [1] is a test-time detection mechanism that flags suspicious inputs, and is conceptually orthogonal to our contribution. In principle, it could be applied on top of any attack (including BadDet+) to mark inputs as unsafe, but it does not remove or repair the underlying backdoor. Analysing the interaction between BadDet+ and TRACE, e.g., adaptive attacks versus detector thresholds, is a valuable direction, but it amounts to a new line of work focused on attack-defence interplay, rather than on attack formulation and evaluation alone.
> > >
> > > To make this clear to readers, we have sharpened the wording in the paper so that our robustness claims are explicitly stated as relative to prior object-detection backdoor attacks under FT/FT-SAM, and we now list pruning-based defences, test-time detectors such as TRACE, and input transformations (JPEG, noise, etc.) as out-of-scope defences that we see as natural targets for a follow-on, defence-oriented study.
> > >
> > > (2) On visual demonstrations
> > >
> > > We apologise if our earlier changes were not clearly signposted. In our initial rebuttal to the comment “Lack of visual demonstrations. I am curious about the proposed backdoor attack's visual demonstrations, which are more direct to show its effectiveness.” we wrote:
> > >
> > > “In response to this comment and Comment 1.2, we have added more detailed descriptions and visualizations of the triggers used in our experiments. We also compare the performance of BadDet+ when different triggers are used in Appendix A.4.
> > >
> > > In addition, as part of our response to Comment 1.1, we have included several further inference examples for each method beyond those shown in Figure 1. These qualitative examples, now provided in Appendix A.9, more directly illustrate the effect of BadDet+.”
> > >
> > > We will ensure that these locations are explicitly referenced in the final version so that readers can easily find the visual demonstrations.

---

> > > > ### Author Response · Authors · 2025-11-24
> > > >
> > > > (3) On intuition and design of the loss
> > > >
> > > > Thank you for clarifying the concern. The design of the BadDet+ loss is guided by two empirical observations and one modelling choice:
> > > >
> > > > 1. Existing-object focus. In both RMA and ODA, the main failure mode of existing attacks arises from the detector continuing to assign high confidence to the original class on trigger-bearing objects (duplicate detections, incomplete disappearance), rather than from a lack of target-class capacity.
> > > >
> > > > 2. Thresholded behaviour. Whether a prediction survives as a detection is governed by confidence thresholds; what matters is not only the ranking of logits, but whether the original-class logit lies above a decision boundary.
> > > >
> > > > 3. Unifying RMA and ODA. Treating background as a special “target class’’ allows ODA to be viewed as a special case of RMA. This suggests that a single mechanism should suppress the original-class logit on triggered boxes and then let the standard classification loss decide whether to redirect to a target class (RMA) or to background (ODA).
> > > >
> > > > From these observations, the design goal is to impose a soft constraint that (i) activates only on trigger-bearing objects and (ii) sharply penalises original-class logits that exceed a chosen confidence boundary, while remaining essentially inactive when the logit is below that boundary. This leads directly to the softplus / log-barrier form around a threshold $\tau$, which can be interpreted as a smooth approximation of the hard constraint “do not be confidently correct on the original class when the trigger is present”.
> > > >
> > > > In the revised text, we have added a short paragraph in Section 4, before the formal definition, that explains this design rationale explicitly so that the intuition precedes the mathematical formulation rather than only appearing afterwards. Appendix A.7 then develops the more formal perspective.

---

> > > ### Author Response · Authors · 2025-11-25
> > >
> > > To address the concerns regarding our evaluation of potential defense strategies, we have conducted an additional experiment evaluating BadDet+ under JPEG compression as an input-sanitisation step. Specifically, we examined whether applying JPEG compression to test images can meaningfully weaken BadDet+. The full results are provided in a new Appendix (A.9), with forward references added where relevant to the updated manuscript. For convenience, we summarise the main findings here.
> > >
> > > To evaluate the robustness of BadDet+ to simple input sanitisation, we measured the impact of varying JPEG compression quality settings (lower quality = stronger compression) on the performance of the backdoored model. Because JPEG compression is lossy, this experiment provides a direct indication of how brittle BadDet+ is to low-cost, test-time transformation. In Figs. 9(a) and 9(b), we report results across multiple JPEG quality settings for the FCOS and DINO model architectures, under both RMA and ODA.
> > >
> > > Overall, we do not find any compression level that substantially reduces ASR@50 without also causing significant degradation in mAP for either RMA or ODA. For the RMA setting (Fig. 9(a)), the most favourable configuration (JPEG quality of 25) still yields $\text{ASR@50}>0.6$ in all cases while keeping $\text{TDR@50}<0.45$. The ODA results in Fig. 9(b) exhibit a similarly limited reduction in ASR@50 across the tested quality levels.
> > >
> > > These findings indicate that simple JPEG-based sanitisation cannot neutralise the threat posed by BadDet+, although it can attenuate the backdoor effect to some extent, particularly at compression levels that already start to harm clean performance. While we only evaluate JPEG compression explicitly, the observed trade-off (modest ASR reduction at the cost of substantial mAP degradation) is characteristic of aggressive test-time transformations that heavily perturb the input. This suggests that low-cost transformation-based sanitisation alone is unlikely to fully neutralise robust OD backdoors without impacting clean performance, although a systematic study of other transforms (e.g., noise, blur, diffusion-based purification) remains an interesting direction for future, more defence-focused work.
> > >
> > > | Quality      | mAP | ASR |
> > > | ------------ | ------- | ------- |
> > > | **Baseline** | 0.5482  | 0.8511  |
> > > | 10           | 0.3209  | 0.6651  |
> > > | 25           | 0.5117  | 0.6044  |
> > > | 50           | 0.5429  | 0.6405  |
> > > | 75           | 0.5590  | 0.6908  |
> > > | 100          | 0.5680  | 0.7570  |
> > > FCOS - ODA
> > >
> > > | Quality      | ODA-mAP | ODA-ASR |
> > > | ------------ | ------- | ------- |
> > > | **Baseline** | 0.5432  | 0.9369  |
> > > | 10           | 0.2332  | 0.7717  |
> > > | 25           | 0.4639  | 0.7257  |
> > > | 50           | 0.5161  | 0.7614  |
> > > | 75           | 0.5448  | 0.8862  |
> > > | 100          | 0.5448  | 0.8862  |
> > > DINO - ODA
> > >
> > > | Quality      | RMA-mAP | RMA-ASR | RMA-TDR |
> > > | ------------ | ------- | ------- | ------- |
> > > | **Baseline** | 0.5582  | 0.9343  | 0.1474  |
> > > | 10           | 0.2978  | 0.5988  | 0.3473  |
> > > | 25           | 0.5124  | 0.7151  | 0.4187  |
> > > | 50           | 0.5624  | 0.8122  | 0.3204  |
> > > | 75           | 0.5624  | 0.8122  | 0.3204  |
> > > | 100          | 0.5708  | 0.8815  | 0.2451  |
> > > FCOS - RMA
> > >
> > > | Quality      | RMA-mAP | RMA-ASR | RMA-TDR |
> > > | ------------ | ------- | ------- | ------- |
> > > | **Baseline** | 0.5335  | 0.9218  | 0.0515  |
> > > | 10           | 0.2127  | 0.3939  | 0.2391  |
> > > | 25           | 0.4668  | 0.6356  | 0.2605  |
> > > | 50           | 0.5164  | 0.7083  | 0.2173  |
> > > | 75           | 0.5300  | 0.7716  | 0.1664  |
> > > | 100          | 0.5490  | 0.8601  | 0.0877  |
> > > DINO - RMA

---

### Official Review · Reviewer_mYnH · 2025-10-20

**Soundness:** 3
**Presentation:** 3
**Contribution:** 3
**Rating:** 6
**Confidence:** 3

**Summary:**

This paper argues that existing backdoor attacks in object detection are inadequate in terms of inconsistent evaluation and/or unrealistic assumptions. They propose a method that unifies RMA and ODA attacks with a mechanism that suppresses true class predictions. Experiments demonstrate generally improved performance compared to prior approaches.

**Strengths:**

The paper argues that there are issues with current evaluation approaches, e.g. ASR overstating success and mAP results being skewed by duplicate detections. The proposed TDR is a sound measure to help alleviate some of the issues.

BadDet+ itself is well motivated, and while the underlying idea is simple, unifying untargeted ODA and RMA under a single mechanism is attractive, and the comprehensive experimental results demonstrate its effectiveness.

Overall, this is a good contribution, though perhaps quite incremental.

**Weaknesses:**

As the authors themselves identify, the approach assumes that training is controlled by the adversary. This is a very strong assumption, though it is not unreasonable to assume such a worst case in some scenarios.

The idea is quite simple and incremental on prior work.

Minor:
- The poor performance compared to BadDet on YOLO for MTSD and PTSD is not adequately highlighted in the main text, where it is stated as "on par with BadDet". Some discussion is in the appendix, but this main text mention feels a bit understated.

- The caption on figure 1 reads a bit confusingly

**Questions:**

A couple of minor weaknesses above could perhaps be addressed.

---

> ### Author Response · Authors · 2025-11-20
>
> Thank you for your thoughtful and constructive feedback. We are encouraged by the positive reception of our work and appreciate the time and effort taken to evaluate our paper. In the responses below, we address each comment individually and indicate where clarifications or revisions have been made in the paper. Please let us know if you believe the changes are inadequate or if further clarification is needed. We are eager to work collaboratively with all reviewers during the discussion period to further improve the paper.
>
> **Comment 3.1: Threat Model**
>
> **Response:** In response to Comment 2.1, we have clarified why we adopt this assumption (threat model) and how it relates to data-poisoning scenarios.
>
> Regarding the comparison with existing attacks, we do not claim that they operate under the same assumptions as ours, but rather examine whether existing data-poisoning attacks are reliably embed practically effective backdoors in object detection models. Our analysis indicates that limitations in the evaluation protocols of related prior work have led to overstatement of their effectiveness. After correcting methodological issues in Align and UBA (Align Random and UBA Box in Section 5.3), and evaluating BadDet using TDR@50, we find that their performance falls short of what would be needed to pose a strong practical threat.
>
> We further vary the poisoning ratio for each considered existing attack across different object-detection models (Appendix A.3) and observe that, in almost all cases, we cannot identify a poisoning rate that yields strong ASR@50/TDR@50 while maintaining reasonable mAP. These results suggest that current data-poisoning attacks, even when strengthened, struggle to robustly satisfy the backdoor objective in object detection.
>
> This motivates our choice to study a stronger threat model: if existing data-poisoning attacks are insufficient, it is important to understand whether a more powerful adversary, one who is able to influence the training objective, can succeed. This rationale is consistent with established backdoor literature in image classification, where attackers are often assumed to be capable of modifying the training pipeline, and is further justified by common practices of outsourcing training (e.g., to cloud service providers) or relying on third-party pretrained model weights.
>
> In light of this comment, as well as Comments 1.3 and 2.1, we have moved key parts of this analysis from Appendix A.3 into the main text to make the motivation for our assumed threat model more prominent.

---

> ### Author Response · Authors · 2025-11-20
>
> **Comment 3.2: Incremental**
>
> **Response:** We acknowledge the reviewer’s concern that the core idea behind the work may appear simple and incremental. However, we would like to highlight several aspects of our work that we believe provide non-trivial contributions, particularly in the attack formulation, evaluation protocol, and critique of existing methods. We summarize our contributions as follows.
>
> - **Diagnosing existing evaluation blind spots.** We reveal that common OD backdoor evaluations systematically miss duplicate/phantom detections and geometry sensitivity, and we make these effects measurable with detector-appropriate metrics.
> - **Evaluation protocol tailored to OD backdoors.** We design an evaluation recipe that explicitly probes position/scale sensitivity via trigger sweeps, assesses digital-to-physical transfer, and enforces consistent ASR/clean-mAP reporting across both ODA and RMA settings.
> - **Ruling out simple fixes.** We demonstrate that straightforward modifications to prior attacks (e.g., box-assignment tweaks, aligned random sizing, and increased poisoning ratios) do not resolve these failure modes, as confirmed by our targeted ablations.
> - **Unified attack mechanism.** We introduce a log-barrier penalty that integrates seamlessly with both RMA and ODA and reconciles head-logit semantics across architectures (independent sigmoid vs. softmax), yielding a detector-agnostic objective that preserves clean performance while reliably inducing the backdoor behavior.
>
> Regarding the proposed approach, we emphasize that it includes several non-trivial conceptual elements beyond a "simple tweak." First, we show that ODA is actually a special case of RMA, which allows us to apply the same penalty in both settings and to interpret them within a unified framework. We have further clarified this connection in response to Comment 2.2. Second, we demonstrate how the proposed penalty can be adapted to yield a softmax-compatible formulation that remains faithful to each detector’s native head design. Third, as shown in Section A.5, the proposed penalty does not interfere with the detector’s existing training objectives, which is crucial for preserving clean performance while inducing reliable backdoor activation.
>
> Finally, we note that, even aside from the specific penalty, our systematic analysis of existing evaluation protocols and attack variants provides actionable insights with important implications for future work. In particular, by (i) exposing and quantifying previously overlooked failure modes and (ii) empirically ruling out a range of intuitive "simple fixes," our study clarifies which design choices are actually necessary to obtain robust and physically reliable OD backdoors, rather than offering marginal improvements within existing imperfect evaluation setups.
>
> In light of this comment and Comment 2.3, we have revised the contributions paragraph in Section 1 to better reflect the points made in our responses.
>
> **Comment 3.3: Cases where BadDet+ underperforms BadDet**
>
> **Response:** We agree that the main text should more clearly highlight the cases where BadDet+ underperforms BadDet on YOLO for MTSD and PTSD. In the revised manuscript, we have updated the YOLO results paragraph to explicitly acknowledge this gap. Specifically, we now state that, on YOLO with MTSD and PTSD, BadDet+ lags behind BadDet in terms of ASR/TDR while maintaining comparable clean mAP, and we point the reader to the detailed discussion in the appendix. This makes the performance difference more visible in the main text and aligns the narrative with the quantitative results.
>
> **Comment 3.4: Figure 1 Caption**
>
> **Response:** We have modified Figure 1's caption to more clearly explain the depicted scenario and the role of each panel, with the aim of improving readability and reducing ambiguity.

---

> ### Comment · Reviewer_mYnH · 2025-11-20
>
> Thank you for your responses. The updates proposed should improve the paper somewhat in terms of clarity and exposition. That said, I shall maintain my original score.

---

> ### Author Response · Authors · 2025-11-21
>
> Thank you again for taking the time to review our paper and to read our responses. We have incorporated the suggested clarifications and revisions into the updated manuscript, so that, if the paper is accepted, the camera-ready version will directly reflect them.
>
> If you have not yet had the opportunity to look at the revised version, we would be very grateful if you could briefly indicate whether these changes address your main concerns, or if there remain specific aspects that you still find unsatisfactory or unclear. Even a short comment would be extremely helpful for us in understanding the key blocking issues and in improving this line of work, regardless of the final decision.

---

> > ### Comment · Reviewer_mYnH · 2025-11-24
> >
> > Thank you. I have viewed the revisions and the exchanges with other reviewers, and the paper is certainly clearer as a consequence. The clarification of the contributions addresses directly one of my main concerns that it is somewhat incremental. As such, I am motivated to revise my score.

---

> > > ### Author Response · Authors · 2025-11-25
> > >
> > > We want to thank you for reviewing our changes. The feedback from you and the other reviewers has led to substantive improvements that we believe will help the work be better received at ICLR (if accepted) or at another venue.

---

### Official Review · Reviewer_Nubo · 2025-10-24

**Soundness:** 3
**Presentation:** 2
**Contribution:** 2
**Rating:** 2
**Confidence:** 4

**Summary:**

This paper introduces BadDet+, a penalty-based backdoor attack framework for object detection, unifying region misclassification and object disappearance with log-barrier penalties, achieving strong physical robustness and transferability while maintaining clean accuracy, revealing critical security vulnerabilities in detection models.

**Strengths:**

1. The authors provide a clear and comprehensive related work.
2. The authors identify the incomplete success of traditional backdoor attacks and propose a new loss function to optimize these bad cases, making the backdoor more robust and effective.

**Weaknesses:**

1. The paper actually adopts a stronger attacker assumption — the ability to manipulate the loss function (i.e., the training process) — which is clearly different from traditional data poisoning attacks. Although I acknowledge the validity of this threat model, I believe the authors should clearly explain these points before introducing their method. Otherwise, comparing it with other data poisoning–based backdoor attacks is, in my view, of limited significance.
2. The paper does not clearly explain in the methodology section how it becomes “a single formulation that generalizes to RMA and untargeted ODA settings.” I believe providing concrete examples would help make this point much clearer.

**Questions:**

1. I’m still not quite clear on how Section 4.1 FORMULATION demonstrates that ODA and RMA can be unified under a single framework — I couldn’t find a clear explanation of this point. Also, what if more types of attacks are considered? Can they also be incorporated into this unified framework? In other words, why are ODA and RMA specifically chosen?

2. Why does the paper claim that BadDet+ bridges the synthetic-to-physical performance gap and achieves position- and scale-invariant backdoor behavior? These benefits don’t seem to come from the core method described in Section 4.1 but rather from common data augmentation techniques. If that’s the case, any backdoor attack could be similarly enhanced. In other words, these advantages are not intrinsic to the proposed method itself.

---

> ### Author Response · Authors · 2025-11-20
>
> Thank you for your thoughtful and constructive feedback. In the responses below, we address each comment individually and indicate where clarifications or revisions have been made in the paper. Please let us know if you believe the changes are inadequate or if further clarification is needed. We are eager to work collaboratively with all reviewers during the discussion period to further improve the paper.
>
> **Comment 2.1: Threat Model**
>
> **Response:** We acknowledge the concerns regarding the assumed threat model. In the second paragraph of Section 4, before introducing our approach, we state that our attack model is stronger than in existing data-poisoning attacks, since the adversary can influence the training objective.
>
> In the revised manuscript, we make this more explicit by adding a dedicated Threat model paragraph to Section 4 that contrasts our setting with traditional data-poisoning scenarios and explains why we study this stronger attacker.
>
> Regarding the comparison with data-poisoning-based attacks, our goal is not to claim that these methods operate under the same assumptions as ours, but rather to examine whether existing data-poisoning attacks can reliably embed practically effective backdoors in object detection models. Our analysis indicates that limitations in the evaluation protocols of related prior work have led to overstatement of their effectiveness. After correcting methodological issues in Align and UBA (Align Random and UBA Box in Section 5.3) and evaluating BadDet using TDR@50, we find that their performance falls short of what would be needed to pose a strong practical threat.
>
> We further vary the poisoning ratio for each considered existing attack across different object-detection models (Appendix A.3) and observe that, in almost all cases, we cannot identify a poisoning rate that simultaneously achieves strong ASR@50/TDR@50 and maintains reasonable mAP. These results suggest that current data-poisoning attacks, even in their most aggressive configurations, struggle to robustly satisfy the backdoor objective in object detection.
>
> This finding motivates our choice to study a stronger threat model: if existing data-poisoning attacks are insufficient, it is important to understand whether a more powerful adversary, one who can influence the training objective, can succeed. This justification is also consistent with established backdoor literature in image classification, where attackers are often assumed to be able to modify the training pipeline, and is further supported by current practice, in which training is frequently outsourced (e.g., to cloud service providers) or relies on third-party pretrained model weights.
>
> In light of this comment, as well as Comments 1.3 and 3.1, we have (i) introduced the dedicated Threat model paragraph before presenting our method and (ii) moved key parts of the above analysis from Appendix A.3 into the main text, in order to make the justification for our assumed threat model and the significance of the comparison to data-poisoning attacks more prominent.

---

> ### Author Response · Authors · 2025-11-20
>
> **Comment 2.2: Clarification of Single Formulation**
>
> **Response:** We agree that our exposition in Section 4 can better highlight how the proposed formulation unifies RMA and untargeted ODA attacks. In the revised version, we have updated Section 4.1 to explicitly signpost this connection and to include concrete examples. For convenience, we summarise the reasoning here.
>
> Although RMA and ODA backdoor attacks are usually described as having different goals, the design of modern object detectors makes them closely related. In both cases, successful attack must (i) suppress predictions that confidently assign the original class to trigger-bearing objects and have high IoU with the corresponding ground-truth box, and (ii) replace these predictions either with a target-class prediction (RMA) or with no prediction (ODA). Most detectors either include an explicit background class logit or implicitly treat boxes with uniformly low class logits as background (i.e., no object). Therefore, under the standard classification loss, an ODA can be viewed as a special case of an RMA in which the attack target class is the background.
>
> Our formulation makes this connection explicit. We introduce a log-barrier penalty that is activated whenever two conditions hold: (i) a predicted box has high overlap with a ground-truth object containing the trigger and (ii) the prediction assigns high confidence to that object’s original class. Penalizing such cases systematically discourages confident original-class predictions on trigger-bearing objects. The standard classification objective then naturally steers the model towards predicting either the target class (RMA) or the background class (ODA). When viewed together, this shows that our formulation provides a single mechanism that facilitates both behaviors, with ODA arising as a special case that does not require any additional modification.
>
> Regarding additional attack types, our framework can, in principle, incorporate other backdoor objectives that act on predictions for existing trigger-bearing objects (e.g., alternative object-disappearance or misclassification behaviors). In contrast, backdoor attacks that aim to generate new objects in previously empty regions (object-generation attacks) are not compatible with this design, as they require modeling qualitatively different mechanisms. For these reasons, and because RMA and untargeted ODA are currently the two dominant backdoor attack types in the object-detection literature, we focus on RMA and untargeted ODA in this work.
>
> **Comment 2.3: Clarification of Claims**
>
> **Response:** Our claims about bridging the synthetic-to-physical performance gap and achieving position- and scale-robust behavior are based on the empirical results in Section 5.3. In particular, the small performance differences between MTSD and PTSD for BadDet+ in Tables 3 and 4, especially when compared to other methods, indicate that BadDet+ transfers more reliably from the synthetic triggers used during training to real triggers placed on physical objects. Similarly, the small performance differences between the Fixed and Random trigger settings for BadDet+ (again relative to other methods) support our claim that it is less sensitive to variations in trigger position and scale.
>
> We agree that these properties do not arise from the log-barrier term alone. However, BadDet+ uses exactly the same detector-specific data-augmentation pipelines as the baselines and does not use any additional bespoke augmentation. In other words, all attacks operate under comparable augmentation regimes, yet only BadDet+ consistently exhibits strong synthetic-to-physical transfer and reduced sensitivity to trigger position and scale. This suggests that BadDet+ causes the model to internalize the backdoor behavior more robustly compared to attacks that rely solely on data poisoning.
>
> To reduce potential confusion, we have revised the stated contributions in Section 1 to emphasize the broader evaluation and analysis that we provide. We have also presented and discussed the robustness properties (synthetic-to-physical transfer and position/scale robustness) as empirical findings in the results section.

---

> > ### Comment · Reviewer_Nubo · 2025-11-20
> >
> > Thank you for the authors’ response. I have carefully gone through the revision. First, I am very satisfied with the revised description of the threat model; it is now much clearer.
> >
> > Regarding Comment 2.3, I still do not see a direct insight. Can it be understood at a high level that, under a loss-manipulation paradigm like BadDet+, the data augmentation paradigm leads to a more-than-additive (1+1>2) improvement in performance?
> >
> > For Comment 2.2, I think the points made by the authors are fully consistent with my understanding, which is very good. However, a strong paper usually distills a fairly general framework. For example, I found that [1] and [2] both try to categorize attacks into two classes and then design corresponding defenses. Although the authors explicitly state that “we focus on RMA and untargeted ODA in this work”, I still consider this to be a potential weakness compared with [1] and [2].
> >
> > Overall, I am satisfied with the authors’ revisions and will increase my score, but due to some inherent limitations, I believe the paper can only remain within this score range.
> >
> > [1] ODSCAN: Backdoor Scanning for Object Detection Models. S&P 24.
> > [2] Test-time backdoor detection for object detection models. CVPR 25.

---

> > > ### Author Response · Authors · 2025-11-21
> > >
> > > We thank you for carefully reading our response. We are encouraged by your positive response to most of our revisions and appreciate the additional comments you have provided. While we understand that further clarification may not substantially change your overall evaluation, we hope that the following points help address your remaining concerns, particularly regarding Comments 2.2 and 2.3.
> > >
> > > **Comment 2.4:** Regarding Comment 2.3, I still do not see a direct insight. Can it be understood at a high level that, under a loss-manipulation paradigm like BadDet+, the data augmentation paradigm leads to a more-than-additive (1+1>2) improvement in performance?
> > >
> > > **Response:** In this context, we understand your use of "data augmentation" as referring to the standard data-poisoning paradigm (i.e., augmenting the training set with patched and relabeled samples), rather than to basic geometric or photometric transformations such as horizontal flips. Our initial response implicitly focused on the latter, so we clarify the intended intuition here.
> > >
> > > At a high level, existing backdoor attacks already follow a common pattern: a fraction of training objects are poisoned (patched and relabeled), and the model is trained with a standard detection loss. Our experiments show that, under this paradigm, the learned backdoor representations are often brittle in practice, especially for ODA-based methods: they struggle to (i) achieve high ASR for ODA or low TDR for RMA, (ii) transfer reliably when triggers are physically placed on real objects, and (iii) remain robust when the trigger is scaled and arbitrarily placed within the bounding box. These issues persist even when the poisoning ratio is pushed to high values as increasing the poisoning rate tends to improve ASR or TDR at the cost of degrading mAP, indicating that data poisoning together with a standard loss alone typically fails to find minima that satisfy both the backdoor and clean objectives simultaneously (see Appendix A.3 and Section 5.3).
> > >
> > > BadDet+ addresses this by introducing a log-barrier penalty term to the learning objective that is active only on poisoned samples and explicitly penalizes confident original-class predictions on trigger-bearing objects. This additional constraint steers optimization towards minima where the backdoor objective is robustly satisfied on poisoned samples, while clean-task performance is preserved, thereby obviating the need to trade off mAP to effectively induce the backdoor behavior.
> > >
> > > From this perspective, the effect is less about a "1+1>2" additive gain of "loss manipulation + poisoning," and more akin to a constrained optimization viewpoint: BadDet+ effectively optimizes a modified objective with an extra constraint that is enforced on poisoned examples, rather than relying on standard poisoning and training dynamics to implicitly discover and encode such behavior. We add a brief clarification along these lines to the conclusion section to make this intuition more explicit.
> > >
> > > **Comment 2.5:** For Comment 2.2, I think the points made by the authors are fully consistent with my understanding, which is very good. However, a strong paper usually distills a fairly general framework. For example, I found that [1] and [2] both try to categorize attacks into two classes and then design corresponding defenses. Although the authors explicitly state that “we focus on RMA and untargeted ODA in this work”, I still consider this to be a potential weakness compared with [1] and [2].
> > >
> > > **Response:** We are glad that our earlier response clarified our position regarding Comment 2.2. We agree that our focus on a non-exhaustive attack set can be viewed as a limitation in terms of breadth, and we now make this explicit in the revised conclusion and threat-model discussion. Our intention in this work is to go deeper into a specific and practically important family of attacks, rather than to propose a taxonomy that spans all attack behaviors such as those considered in the defensive works of [1] and [2]. In particular, BadDet+ is designed for attacks that manipulate predictions for existing objects (RMA and untargeted ODA), where our analysis reveals systematic weaknesses in prior work: duplicate detections, geometry sensitivity, and poor synthetic-to-physical transfer under realistic training pipelines.
> > >
> > > However, we would like to also clarify that among attacks covered in [1] and [2], the only main type not targeted in our formulation is the object generation attack (OGA), which induces detections in previously empty regions rather than modifying predictions for annotated objects. In this regime, the original BadDet already proposes an effective OGA with a sound evaluation, achieving ASR > 90 in almost all tested cases. Subsequently, we do not observe the same systematic failure modes that motivate BadDet+ for RMA/ODA, hence, we did not identify a compelling need to redesign OGA within our framework.

---

> > > > ### Comment · Reviewer_Nubo · 2025-11-21
> > > >
> > > > Thank you to the authors for the clarification. Regarding Comment 2.4, I understand that it essentially amounts to a constrained optimization, or a weighted treatment. In theory this should indeed lead to improvements, but the design itself is not particularly elegant. For Comment 2.5, I maintain my original assessment.

---

> ### Author Response · Authors · 2025-11-24
>
> Thank you for the further clarification and for updating your score during the discussion. We understand and respect your perspective on the design choice in BadDet+. Our goal in this work was to prioritise a mechanism that is simple and practically usable across multiple modern detectors (Faster R-CNN, FCOS, DINO, YOLO) without architecture-specific modifications, even if it is not the most theoretically elegant possible formulation.
>
> We have tried to be explicit in the paper about the scope of our contribution: we focus on RMA and untargeted ODA, highlight systematic weaknesses in existing attacks under our proposed evaluation protocol, and introduce a training loss augmented with a log-barrier penalty term that induces a constrained-optimisation-style behaviour on poisoned samples to address those specific failure modes. We also clearly acknowledge the limitations in breadth, and we outline opportunities for more general frameworks and defence-centric studies as important directions for future work.
>
> We appreciate the time and care you have put into reviewing and discussing the paper, and we thank you for engaging with us throughout the rebuttal process. The feedback from you and the other reviewers has led to substantive improvements, which we believe will help the work be more clearly understood and better received by the community.

---

### Official Review · Reviewer_Pccc · 2025-10-25

**Soundness:** 3
**Presentation:** 3
**Contribution:** 3
**Rating:** 6
**Confidence:** 4

**Summary:**

The paper proposes novel backdoor poisoning attack BadDet+, which includes a loss into the training objective that penalizes the appearance of the true label in top classes (at prediction time).
Using this regularization, the methodology ensures that the scores of the true label drops in presence of the trigger.
Also, the paper proposes novel evaluation metrics, since other attacks could still exhibit the real label as one of the most likely class during prediction, and this is confirmed by the experiments.

**Strengths:**

+ the simplicity of a training time regularizer makes this technology easy to understand and also to stage
+ the paper also propose an ablation study on the parameters of the methodology, highlighting the depth of the study
+ novel metrics show that previous work might have overfitted the goal of mislabelling rather than being sure that the true label is not considered

**Weaknesses:**

- the fact that the paper is presenting a new metric, and with this the performances of the proposed methods are way way better than the literature might rise the doubt on the metric being overfit by BadDet+. Hence, it would be better to show some examples also on regular metrics, or ablation studies also on the 0.5 that has been deemed threshold on the IoU.
- there is no clear technical description of the nature of the trigger, could its shape and color change the entire method? Like triggers that, by chance, are similar to the object on which they are applied, thus being less effective. The paper should better discuss how scenarios like these were avoided.
- even with a different metric, it is different to compare results: were the methods trained in the same settings? Same poisoning ratios? Same trigger sizes? Results might change a lot, considering that the paper states that default parameters have been used for the techniques of state of the art. The appendix provides some insights, but they are not incredibly clear and should be moved to the main paper to some extent.
- limitations are not discussed, the paper draws conclusions without considering possible issues of the proposed approach.

**Questions:**

1) what happens in same backdooring conditions, i.e. same ratio of poisoning samples, same trigger sizes, etc?
2) what happens whether the metrics AS@50 changes ratio? Like 10 to 90? How this is can be connected to the standard ASR when varying these quantities?

---

> ### Author Response · Authors · 2025-11-20
>
> Thank you for your thoughtful and constructive feedback. We appreciate the time and effort taken to evaluate our paper. In the responses below, we address each comment individually and indicate where clarifications or revisions have been made in the paper. Please let us know if you believe the changes are inadequate or if further clarification is needed. We are eager to work collaboratively with all reviewers during the discussion period to further improve the paper.
>
> **Comment 1.1: Metrics**
>
> **Response:** This comment raises three related points concerning our use of standard metrics, the introduction of TDR@50, and the choice of IoU threshold. We address each in turn.
>
> *Existing metrics:* While our work identifies limitations in prior evaluations and proposes a more systematic evaluation protocol, we intentionally build on existing practice and extend it rather than introducing an entirely new metric suite, so that our results remain comparable to prior work. In particular, ASR@50 is already used in both BadDet and Align. In our evaluations, we follow this convention and reports ASR@50 and mAP for all methods, hence the performance gaps we observe are not an artifact of introducing TDR@50. One important consideration is that, although Align reports ASR@50, its implementation does not scale the trigger with respect to object size, unlike BadDet and UBA. Scaling the trigger with the object is a natural design choice, since a physical trigger placed on a distant object appears smaller than on a nearby one.
>
> To test the limitations of existing evaluation protocols more explicitly, we added an ablation (Appendix A.2.2-A.2.3) that re-evaluates Align and UBA under their original settings. For UBA, we find that Poison mAP can be low across very different ASR@50 regimes (low for Faster R-CNN and FCOS, high for DINO), whereas BadDet+ consistently achieves low Poison mAP and high ASR@50. This indicates that Poison mAP, as originally proposed by UBA, is not a reliable indicator of backdoor attack success, and that ASR (as used in BadDet and Align) is a more appropriate measure of attack success. For convenience, we provide both results in a seperate response (see UBA Evaluation).
>
> For Align, we compare fixed-size versus scaled triggers and observe up to a 23.58% ASR gap. When using randomly scaled triggers (Align Random), this gap shrinks to 2.18%. This shows that properly accounting for trigger scaling, as in BadDet and UBA, substantially affects Align’s performance. In addition, in response to Comment 4.2, we have included further qualitative examples in Appendix A.9 to illustrate failure cases of existing methods. For convenience, we provide both results in a seperate response (see Align Evaluation).
>
> *New metric (TDR@50):* To our knowledge, we are the first to introduce TDR@50 for object detection. However, the underlying idea is aligned with existing practice in adjacent contexts. For example, in image classification, recovery accuracy (RA) is commonly used to measure whether backdoored inputs are correctly classified after mitigation. Tables 2 and 4 show that TDR is essential for revealing the duplicate-detection behavior discussed in Section 3, which standard metrics alone largely fail to capture. We have clarified this connection in Section 5.2 and emphasize that TDR is used in addition to, not in place of, standard metrics such as ASR@50 and mAP.
>
> *IoU range:* We agree that the IoU choice can affect ASR and TDR in RMA. To address this, we have added an ablation study varying the IoU threshold in Appendix A.2.1. We find that ASR and TDR are largely stable up to IoU=0.8, after which both begin to decrease. For ODA, increasing IoU beyond 0.5 only increases ASR, since disappearance becomes easier to satisfy under stricter IoU matching. Therefore, for consistency with existing work, we retain IoU=0.5 as the main setting in the paper and refer the reader to Appendix A.2.1 for the full analysis. For convenience, we provide both results in a seperate response (see ASR and TDR IoU Evaluation).
>
> Overall, our experiments indicate that BadDet+ does not simply "overfit" to TDR@50. It performs favorably under standard object-detection backdoor metrics (ASR@50 and mAP), while TDR@50 and the IoU ablations provide a more nuanced view of backdoor behavior rather than overstating BadDet+’s performance.

---

> ### Author Response · Authors · 2025-11-20
>
> **Comment 1.2: Trigger Evaluation**
>
> **Response:** The trigger description in Section 5.1 was indeed underspecified. We have now clarified that all experiments use a square patch trigger, with a blue square as the default. This choice matches the trigger used in the PTSD real-world dataset and facilitates a direct comparison between synthetic (MTSD) and physical (PTSD) domains.
>
> To address the concern that trigger appearance might affect the approach’s effectiveness, we have added an ablation varying both color and visual complexity. Specifically, we consider (a) green, (b) red, (c) yellow, and (d) multi-colored square triggers. Red and yellow are chosen to more directly probe scenarios where the trigger may resemble underlying objects. Across all variants, we observe very similar mAP, ASR@50, and TDR@50, indicating that our conclusions do not hinge on a particular trigger color or visual pattern. Notably, the original blue trigger yields the lowest ASR@50 among the tested variants. Full results and analysis are provided in Appendix A.4. We include the corresponding results table below for convenience.
>
> **Comment 1.3: Evaluation Procedure**
>
> **Response:** We recognize that fair comparison requires aligning training and attack settings, and we have clarified this more explicitly in the paper. For training, we use the standard pipelines and hyperparameters provided with each architecture, i.e., the official torchvision scripts for Faster~R-CNN and FCOS, and the original Facebook Research and Ultralytics repositories for DINO and YOLOv5.
>
> For the attacks, we adopt the poisoning ratios and trigger-application strategies proposed in the original papers and apply them consistently across methods wherever possible. The only structural exception is Align, which uses a fixed-size trigger placed multiple times in each image, rather than scaling the trigger with the object size. By contrast, BadDet, UBA, and Morph employ object-scaled triggers. In these methods, the poisoning rate is the only tunable attack parameter. As shown in Appendix A.3, increasing poisoning ratio beyond the original values does not materially change our conclusions. To make these settings and their impact more transparent, we have revised Section 5.1 and moved key poisoning-ratio results from Appendix A.3 to the main text (see the "Poisoning Ratio" paragraph in Section 5.3).
>
> **Comment 1.4: Limitations**
>
> **Response:** In the revised manuscript, we now make the limitations of BadDet+ explicit in the conclusion section. While we already note that the current lack of object-detection-specific mitigation strategies poses a significant risk, given the effectiveness of BadDet+, this is a limitation of the broader ecosystem rather than of our work. We also state at the beginning of Section 4 that our approach assumes a stronger threat model than existing data-poisoning–based attacks, and, in Section 5.3, we discuss cases (e.g., YOLO on MTSD/PTSD) where BadDet+ does not improve over BadDet.
>
> We have now consolidated and highlighted these points in the conclusion section as explicit limitations of our work, clarifying the scope of our contributions and the conditions under which our conclusions are intended to hold.

---

> ### Author Response · Authors · 2025-11-20
>
> **UBA Evaluation**
> |Method|Model|Poison mAP|ASR50|
> |-|-|-|-|
> |UBA|FasterR-CNN|3.75|44.35|
> |UBA|FCOS|4.89|28.65|
> |UBA|DINO|0.19|97.88|
> |BadDet+|FasterR-CNN|0.07|98.46|
> |BadDet+|FCOS|0.26|96.95|
> |BadDet+|DINO|0.86|97.59|
>
>
> **Align Evaluation**
> |Method|Model|Fixed|Scaled|Delta|
> |-|-|-|-|-|
> |Align|FasterR-CNN|61.81|38.23|-23.58|
> |Align|FCOS|55.98|33.35|-22.63|
> |Align|DINO|50.66|32.15|-18.51|
> |Align Random|FasterR-CNN|61.63|61.93|+0.30|
> |Align Random|FCOS|53.05|55.23|+2.18|
> |Align Random|DINO|78.23|79.92|+1.69|
>
>
> **ASR and TDR IoU Evaluation**
>
> COCO
> |Method|ASR@50|TDR@50|ASR@60|TDR@60|ASR@70|TDR@70|ASR@80|TDR@80|ASR@95|TDR@95|
> |-|-|-|-|-|-|-|-|-|-|-|
> |DINO|97.3|1.5|96.1|0.8|94.6|0.4|91.9|0.2|56.1|0.0|
> |FasterR-CNN|99.5|3.2|98.3|0.8|94.9|0.4|87.1|0.3|20.5|0.0|
> |FCOS|99.1|1.5|97.8|0.6|94.5|0.5|85.0|0.3|20.3|0.1|
>
> MTSD
> |Trigger Position|Method|ASR@50|TDR@50|ASR@60|TDR@60|ASR@70|TDR@70|ASR@80|TDR@80|ASR@95|TDR@95|
> |-|-|-|-|-|-|-|-|-|-|-|-|
> |Both|DINO|97.7|1.8|97.5|1.8|96.8|1.8|95.0|1.6|28.2|0.2|
> |Both|FasterR-CNN|98.6|3.2|98.0|3.2|97.9|3.2|93.4|2.3|17.2|0.2|
> |Both|FCOS|97.2|5.7|97.2|5.7|96.3|5.7|93.1|5.3|27.4|0.9|
> |Both|YOLO|94.5|5.2|94.1|5.2|93.6|5.2|90.6|4.8|25.0|0.7|
> |Random|DINO|92.2|5.7|91.8|5.7|91.5|5.5|89.0|5.2|24.2|1.1|
> |Random|FasterR-CNN|97.9|8.2|97.2|8.0|95.4|7.6|88.6|7.1|13.7|1.2|
> |Random|FCOS|92.9|13.3|92.9|13.1|92.4|13.1|87.9|12.1|20.6|3.9|
> |Random|YOLO|88.1|12.8|88.1|12.6|87.4|12.4|83.8|11.9|21.1|1.4|
>
>
> **Additional Trigger Evaluation**
> |Attack|Trigger|FCOS mAP|FCOS ASR@50|FCOS TDR@50|DINO mAP|DINO ASR@50|DINO TDR@50|
> |-|-|-|-|-|-|-|-|
> |RMA|Original|55.86|93.13|16.96|53.46|90.43|5.39|
> |RMA|A|55.10|98.22|5.86|53.15|97.33|0.35|
> |RMA|B|53.04|97.15|7.10|53.47|96.44|2.48|
> |RMA|C|54.43|97.51|6.57|54.82|95.38|1.77|
> |RMA|D|54.00|97.15|9.23|49.34|93.39|3.33|
> |ODA|Original|54.82|83.68|--|54.32|92.31|--|
> |ODA|A|57.09|96.27|--|52.35|99.35|--|
> |ODA|B|55.57|93.04|--|51.55|97.57|--|
> |ODA|C|57.82|96.27|--|54.83|95.49|--|
> |ODA|D|57.46|93.36|--|51.63|95.95|--|
>
> A = Green, B = Red, C = Yellow and D ‎ =  Multi-coloured

---

> > ### Comment · Reviewer_Pccc · 2025-11-24
> > **Thanks for the comments**
> >
> > Thanks for the clarification, I will keep my score since the answers address my comments precisely.

---

> > > ### Author Response · Authors · 2025-11-24
> > >
> > > Thank you for your response and for confirming that our responses have addressed your comments precisely.
> > >
> > > For our own understanding, are there any remaining concerns or limitations that you feel still justify the current overall score, beyond what we have already discussed and revised? If so, we would be grateful if you could briefly highlight them so that we can better position this work and its limitations.
> > >
> > > Conversely, if your overall assessment of the paper has improved after the revisions and discussion, we would, of course, appreciate if that could be reflected in the final numerical score as you see appropriate. In any case, we thank you again for the time and care you have invested in reviewing the paper.

---

### Author Response · Authors · 2025-11-30
**Summary (part 3/3)**

---

### 7. Net effect of the discussion

- Reviewer **Pccc** kept their score at **6** (above the acceptance threshold) and stated that our answers “address my comments precisely.”

- Reviewer **mYnH** increased their score of **6** to **8** and explicitly noted that:
  - the paper is “certainly clearer as a consequence” of the revisions, and
  - the clarified contributions addressed their concern that the work might be incremental, which motivated them to revise their score.

- Reviewer **Nubo** started from a **2** (reject) and **increased** their score after the rebuttal, acknowledging that:
  - the threat model is now much clearer, and
  - our explanations of the unified formulation are consistent with their understanding, while still seeing some limitations in breadth.

- Reviewer **ty81** started from a **0** (strong reject) and **increased** their score after the rebuttal and additional experiments. They still see the paper as below the bar, mainly because they would like to see:
  - a much broader, defence-heavy study (multiple pruning, transformation, and detection defences),
  - more extensive visual analysis in the main text, and
  - an even deeper methodological-intuition section.

  We have improved the robustness claims, added an explicit JPEG experiment, clarified where visual examples are located, and added a pre-formula intuition paragraph in Section 4, but we recognise that these changes may not fully resolve their preferences.

---

In summary, the final version (if accepted) will:

1. Clearly state and justify the stronger, yet realistic, threat model.
2. Provide an explicit unified formulation for RMA and untargeted ODA, with intuition before the formal loss.
3. Clarify the technical role of the log-barrier penalty and its constrained-optimisation flavour.
4. Retain standard metrics while using TDR/IoU studies to reveal failure modes in prior attacks.
5. Make experimental settings (triggers, poisoning ratios, datasets, detectors) transparent and better aligned.
6. Explicitly delimit the scope of robustness claims and position broader defence studies as future work.

We hope this helps the AC quickly see both the contributions and the final state of the paper after rebuttal.

---

### Author Response · Authors · 2025-11-30
**Summary (part 2/3)**

---

### 4. Evaluation protocol, metrics, and “overfitting to TDR@50”

- We keep **standard OD backdoor metrics** (ASR@50 and clean mAP) for all methods, and we do *not* replace them with TDR.

- We introduce **TDR@50** as an *additional* metric, analogous to recovery accuracy in classification backdoor work, to expose:
  - duplicate/phantom detections in RMA (target+original both fire), and
  - cases where ODA’s disappearance is only partial.

- In response to concerns about “overfitting” to TDR@50 (Reviewer Pccc):
  - We added IoU-sweep ablations (ASR/TDR vs IoU threshold) and showed that BadDet+ remains strong up to IoU ≈ 0.8, with degradation only at very high IoU where all methods suffer.
  - We re-evaluated UBA under its original Poison mAP metric and showed that Poison mAP is not a reliable proxy for backdoor success, while BadDet+ still performs well under ASR@50.
  - We re-evaluated Align with fixed vs scaled triggers and showed large ASR gaps when scaling is handled realistically; this motivates our “Align Random” variant.

- Overall, the updated results make clear that BadDet+ performs well under *standard* metrics; TDR and IoU ablations provide a more nuanced view rather than artificially inflating our results.

---

### 5. Experimental clarifications and additions

In response to detailed comments from Reviewers Pccc and ty81, we made the following concrete changes:

- **Trigger specification and sensitivity**
  - Section 5.1 now clearly states that we use a square patch trigger and a blue square by default (matching the PTSD physical trigger).
  - Appendix A.4 adds an ablation over trigger *colour and complexity* (green, red, yellow, multicolour). Performance is similar across variants, and the original blue trigger is actually slightly *weaker*, indicating our conclusions are not tied to a particular design.

- **Poisoning ratios and alignment of settings**
  - We clarified that training uses the standard pipelines from TorchVision (Faster R-CNN, FCOS), FB-Research DINO, and Ultralytics YOLOv5.
  - We aligned poisoning ratios and trigger placement across methods as much as their original designs allow, and we brought the key poisoning-ratio ablation from Appendix A.3 into the main text.

- **Cases where BadDet+ underperforms BadDet**
  - The main text now explicitly calls out the YOLO/MTSD/PTSD cases where BadDet has better ASR/TDR than BadDet+, while clean mAP stays comparable, and refers to the detailed appendix discussion.

- **Visual examples**
  - We added additional qualitative examples and trigger visualisations in the appendix (including extra inference examples for each method), and clarified their location in the text so that readers can easily see the qualitative effects of BadDet+.

---

### 6. Defence evaluation and scope of “robustness”

A major part of Reviewer ty81’s concern was the perceived lack of a broad defence study.

- Our robustness claims are now **explicitly scoped** as:
  *“robustness relative to existing OD backdoor attacks under FT and FT-SAM fine-tuning, not robustness against all possible defences.”*

- We evaluate two generic, architecture-agnostic defences:
  - **Fine-tuning (FT)** with 2-4% clean data, and
  - **FT-SAM**, motivated by recent classification backdoor defence work.

- To address feedback during discussion, we additionally:
  - Ran and reported a **JPEG-compression ablation** as a simple input-sanitisation baseline (new appendix): we found no compression level that substantially reduces ASR@50 without also causing large drops in mAP.

- We explain why we **do not** include, in this attack-focused paper:
  - classification-style pruning defences (which are non-trivial to port fairly to transformer-based detectors like DINO),
  - test-time backdoor detectors such as *“Test-time backdoor detection for object detection models”* (CVPR 2025), which flag suspicious inputs but do not remove the backdoor, and
  - a broad suite of input transformations or diffusion-based purification methods.

  A comprehensive defence benchmark, across four architectures and multiple datasets, would significantly expand the scope and is better suited to follow-up, defence-centric work. We now state this explicitly as a limitation and future direction.

---

### Author Response · Authors · 2025-11-30
**Summary (part 1/3)**

We thank the reviewers for the detailed feedback and the constructive discussion. Below we briefly summarize what the paper does, what changed during rebuttal, and how the final state of the work addresses the main concerns.

---

### 1. Scope and threat model

- The paper revisits backdoor attacks in object detection and shows that existing data-poisoning–based attacks (BadDet, Align, UBA, Morph) are less effective than commonly reported when evaluated under a more disciplined protocol (object-wise ASR@50, TDR@50, position/scale sweeps, and synthetic to physical transfer).

- In response to concerns from Reviewers **Nubo** and **mYnH** about the attacker assumption, Section 4 now has a dedicated **Threat model** paragraph. We explicitly state that:
  - Our threat model is stronger than standard data poisoning, because the adversary can manipulate the training objective (loss).
  - The comparison to data-poisoning attacks is *not* presented as apples-to-apples, but as an empirical check of whether existing attacks can reliably implant practically effective OD backdoors.

- We moved the key poisoning-ratio analysis (previously only in Appendix A.3) into the main text and clarified that, across detectors and poisoning rates, existing attacks struggle to achieve strong ASR@50/TDR@50 without significantly degrading mAP. This motivates studying the stronger, but realistic, threat model where the attacker can also modify the loss (consistent with common practice in image-classification backdoor work and cloud/pretrained-weights scenarios).

---

### 2. Unified formulation for RMA and ODA

- Several reviewers asked how exactly BadDet+ “unifies” RMA and untargeted ODA.

- Section 4.1 has been rewritten to:
  - Make explicit that modern detectors either have a background logit or treat low class logits as background, so **ODA can be viewed as a special case of RMA with background as the target class**.
  - Explain, in words and via concrete examples, that BadDet+ adds a log-barrier penalty which activates only when a prediction both (i) overlaps a trigger-bearing ground-truth box and (ii) assigns high confidence to the original class.
  - Clarify that suppressing the original-class logit on such predictions, and then letting the standard detection loss operate, yields either target-class predictions (RMA) or background (ODA) from the *same* mechanism; no separate ODA-specific machinery is required.

- We also explain that the framework naturally extends to other disappearance/misclassification objectives that act on *existing* objects, and we explicitly delimit that **object-generation attacks (new boxes in empty regions) are out of scope** for this formulation. We note that the original BadDet already provides a strong OGA with ASR > 90% in almost all tested cases and does not exhibit the same systematic failure modes that motivate BadDet+.

---

### 3. What BadDet+ actually contributes (beyond “just a sigmoid”)

One reviewer (ty81) initially characterised the contribution as essentially “using a sigmoid”. We clarified in the paper and in the discussion that the key contribution is the *penalty-based formulation and its integration* into detector training:

- BadDet+ augments the standard detection loss with a **log-barrier penalty** that:
  - operates at **training time** (not only on labels/targets),
  - is applied *only* to predictions overlapping trigger-bearing objects that are still confidently predicted as the original class,
  - has a softplus/log-barrier form centred at a confidence threshold, enforcing a **constrained-optimisation-style behaviour**: “do not be confidently correct on the original class when the trigger is present”.

- This yields a **single training objective** that:
  - unifies RMA and untargeted ODA under one mechanism,
  - is compatible with both independent-sigmoid and softmax heads,
  - preserves clean performance while robustly enforcing the backdoor objective on poisoned samples.

- Appendices A.5 and A.7 are used to develop the intuition and the more formal perspective: how the penalty induces trigger-conditional margin suppression in a feature subspace while leaving clean behaviour largely intact.

- In the conclusion, we now explicitly frame the design as *simple but purposeful*: the goal was a mechanism that is easy to deploy across Faster R-CNN, FCOS, DINO, and YOLO without architecture-specific hacks, rather than maximal theoretical elegance.

---

### Meta-Review · Area_Chair_h3Ts · 2026-01-11

**Summary:**

The paper addresses a critical security gap in object detection (OD) by proposing BadDet+, a unified penalty-based attack framework. The primary concerns raised by reviewers initially included:
* Threat Model Clarity: A concern that the attacker assumption (manipulating the loss function) was significantly stronger than traditional data poisoning.
* Methodological Novelty: One reviewer characterized the technical contribution as "just using a sigmoid".
* Evaluation Robustness: Reviewers questioned the lack of broad defense evaluations (e.g., pruning, input transformations).
* Metric Validity: Doubts regarding whether the proposed TDR@50 metric was merely an artifact designed to make the proposed method look better.

During rebuttal and discussion, the authors substantially clarified the threat model, explicitly reframed robustness claims as relative to existing attacks, expanded explanations of the unified formulation, added ablations (IoU sweep, poisoning ratio, trigger variants, JPEG, SAM), and made limitations explicit. This addressed many of the technical and presentation concerns, though not all philosophical or scope-related reservations (notably from Reviewer ty81 and partially from Reviewer Nubo) were fully resolved.

**Reviewer Concerns:**

Concerns Addressed by Rebuttal
* Threat Model and Scope:*Reviewers Nubo and mYnH were satisfied with the clarified threat model, acknowledging that while strong, it is realistic for outsourced training or pretrained-weight scenarios.
* Metric and Evaluation Protocol: Reviewer Pccc's concerns regarding the TDR@50 metric and IoU thresholds were addressed precisely through IoU-sweep ablations and re-evaluating baselines under standard metrics.
* Trigger Sensitivity: The addition of Appendix A.4 (varying trigger color and complexity) successfully demonstrated that the attack's effectiveness is not tied to a specific trigger design.

⠀Outstanding Concerns
* Breadth of Defense Study: Reviewer ty81 maintained that the paper lacks a "comprehensive" defense study, specifically calling for more complex pruning and diffusion-based purification. The authors argued these are out of scope for an attack-focused paper and non-trivial to port to modern architectures like DINO.

**Reviewer Scores:**

- Reviewer Nubo: 2->4
Explicitly states that "I am satisfied with the authors’ revisions and will increase my score, but due to some inherent limitations, I believe the paper can only remain within this score range."
- Reviewer Pccc: 6->6
Likely would remain at 6. The reviewer explicitly stated that "I will keep my score since the answers address my comments precisely."
- Reviewer mYnH: 6->8
Increased score to 8 after the rebuttal clarified the non-incremental nature of the contributions.
- Reviewer ty81: 0->2
The reviewer acknowledged improvements and raised the score from 0, but still viewed the paper as below par due to scope and novelty concerns.

---

### Decision · Program_Chairs · 2026-01-26

Reject